# Differential controls of MAIT cell effector polarization by mTORC1/mTORC2 via integrating cytokine and costimulatory signals

Huishan Tao [1,4], Yun Pan [1,4], Shuai Chu[1,4], Lei Li [1,4], Jinhai Xie[1], Peng Wang[1], Shimeng Zhang[1], Srija Reddy[1], John W. Sleasman[1] & Xiao-Ping Zhong [1,2,3 ✉]

Mucosal-associated invariant T (MAIT) cells have important functions in immune responses against pathogens and in diseases, but mechanisms controlling MAIT cell development and effector lineage differentiation remain unclear. Here, we report that IL-2/IL-15 receptor β chain and inducible costimulatory (ICOS) not only serve as lineage-specific markers for IFN-γ-producing MAIT1 and IL-17A-producing MAIT17 cells, but are also important for their differentiation, respectively. Both IL-2 and IL-15 induce mTOR activation, T-bet upregulation, and subsequent MAIT cell, especially MAIT1 cell, expansion. By contrast, IL-1β induces more MAIT17 than MAIT1 cells, while IL-23 alone promotes MAIT17 cell proliferation and survival, but synergizes with IL-1β to induce strong MAIT17 cell expansion in an mTOR-dependent manner. Moreover, mTOR is dispensable for early MAIT cell development, yet pivotal for MAIT cell effector differentiation. Our results thus show that mTORC2 integrates signals from ICOS and IL-1βR/IL-23R to exert a crucial role for MAIT17 differentiation, while the IL-2/IL-15R-mTORC1-T-bet axis ensures MAIT1 differentiation.

[1] Department of Pediatrics-Allergy and Immunology, Duke University Medical Center, Durham, NC, USA. [2] Department of Immunology, Duke University Medical Center, Durham, NC, USA. [3] Hematologic Malignancies and Cellular Therapies Program, Duke Cancer Institute, Duke University Medical Center, Durham, NC, USA. [4] These authors contributed equally: Huishan Tao, Yun Pan, Shuai Chu, Lei Li. ✉email: xiaoping.zhong@duke.edu

MAIT cells are innate-like T cells with a restricted TCR repertoire. Although very rare in mice, MAIT cells are the most abundant innate-like T cells in humans, accounting 1–30% of T cells dependent on different organs. They are not only rich in mucosal tissues but also reside in other lymphoid and non-lymphoid organs, suggesting broad roles of these cells in immune responses, tissue homeostasis, and diseases[1–5]. Most MAIT cells express an invariant TCRVα19-Jα33 (iVα19) TCR in mice and an invariant TCRVα7.2-Jα33 TCR in humans, although other TCRα chains can also be used by human and mouse MAIT cells[6–9]. These invariant TCRα chains paired with a limited array of TCR β-chains that are mainly Vβ6 and Vβ8 in mice and Vβ2 and Vβ15 in human[6,9–12]. MAIT cells are restricted to the MHC-I-related molecule MR1 but not to classic MHC molecules expressed on thymocytes[13–15]. MR1 presents microbe-derived riboflavin (vitamin B2) metabolites, several drugs or drug-derivatives, and yet to be identified endogenous ligand(s) but not peptides to MAIT cells[13,14,16–19]. Signal from the iVα19TCR is crucial for MAIT cell development, indicated by enhanced generation of MAIT cells in iVα19TCR transgenic mice and their absence in MR1-deficient mice[8,14,20,21], as well as the importance of commensal bacteria derived 5-(2-oxopropylide-neamino)-6-d-ribitylaminouracil (5-OP-RU) to induce expansion of MAIT precursors and mature MAIT cells[22]. Additionally, iVα19TCR signal plays important roles in TLR- and cytokine-induced MAIT cell activation and MAIT effector lineage polarization in vivo[23–26]. Downstream of the TCR, proximal protein tyrosine kinase Zap70 and the second messenger diacylglycerol (DAG), as well as its tight regulation by DAG kinases α and ζ (DGKα/ζ) are important for MAIT cell development[27,28]. However, additional signal pathways downstream of iVα19TCR and other costimulatory and cytokine signals that regulate MAIT cell development are poorly understood.

It was until the recent development of MR1 tetramers loaded with the riboflavin-derived 5-OP-RU ligand that it became possible to detect MAIT cells in mice[11]. MAIT cells are generated in the thymus from CD4+CD8+ precursors. They mature through three ordered stages: CD24+CD44− stage 1, CD24−CD44− stage 2, and CD24−CD44+ stage 3[29]. Different from conventional αβT (cαβT) cells but similar to iNKT cells, MAIT cells differentiate into multiple effector lineages such as IFN-γ-producing MAIT1 and IL-17A-producing MAIT17 cells in the thymus[11,29,30]. The transcription factors PLZF and SATB1, miR-181a/b-1, Drosha (a member of the microRNA processing machinery), and adapter molecule SAP are critical for MAIT cell functional maturation and differentiation into effector lineages[28–31]. Following agoniztic stimulation, MAIT cells rapidly produce a variety of cytokines and chemokines, such as TNF-α, IFN-γ, IL-17, GM-CSF, IL-10, TGF-β, CCL3, CCL4, and CXCL16, that can regulate innate and adaptive immunity, as well as tissue repair. Some MAIT cells express Granzyme B and display cytotoxic activity[12,32–35]. MAIT cells participate in immune responses against pathogens and tissue repair and abnormal MAIT cells are associated with many diseases, such as infections, cancer, diabetes, systemic lupus erythematosus, and multiple sclerosis[15,25,35–43]. At present, mechanisms that control MAIT cell development and effector lineage differentiation are still poorly understood.

mTOR is a serine/threonine kinase that plays critical roles in diverse cellular processes, such as metabolism, proliferation, growth, autophagy, and differentiation. It signals through two complexes, mTORC1 and mTORC2 that contain essential adapter molecules Raptor and Rictor, respectively[44]. mTOR is activated in T cells following engagement of the TCR, costimulatory molecules, such as ICOS, and multiple cytokines receptors[45–48]. DAG-mediated RasGRP1-Ras-Erk1/2 and PKCθ-Carma1 pathways are important for mTOR activation downstream of the TCR, which is negatively controlled by DGKα/ζ[45,46]. mTOR and its tight regulation by TSC1 and DGKα/ζ ensure proper iNKT-cell generation, effector lineage fate decision, and anti-tumor immunity and many aspects of con-vention αβT cell development and function[47,49–60]. TCR and IL-12 plus IL-18 stimulation can activate mTORC1 in human MAIT cells to promote glycolysis and IFN-γ production[61]. However, the importance of mTOR1 and mTORC2 and their upstream receptors and downstream pathways in MAIT cell development and effector lineage differentiation is unknown.

In this report, we show that CD122 (IL-2/IL-15Rβ) is selectively expressed in MAIT1 cells and is critical for MAIT1 but not MAIT17 cell differentiation by transducing signals from both IL-2 and IL-15 that lead to mTORC1 activation, T-bet upregulation, and subsequent enhancement of MAIT1 cell proliferation and survival. By contrast, ICOS is expressed at high levels in MAIT17 cells. IL-1β, together with IL-23, and ICOS signal through mTORC2 to promote MAIT effector cell, especially MAIT17 cell, differentiation, and expansion. Our data provide new mechanistic insights into MAIT effector differentiation and suggest strategies for selective manipulation of MAIT1/17 cell lineages.

## Results

### IL-2/IL-15Rβ and ICOS as lineage-specific markers for MAIT1 and MAIT17 cells and the crucial role of T-bet for MAIT1 but not MAIT17 differentiation

The frequencies of thymic MAIT cells are very low in mice. We examined thymic MAIT cells after enriching with 5-OP-RU-loaded MR1-tetramers (MR1-Tet) conjugated with either PE or APC and magnetic beads in combination with exclusion of other cell lineages and further defined MAIT cell development stages based CD24 and CD44 expression (Fig. 1a). The costimulatory molecule ICOS and the IL-2/IL-15 receptor β chain (IL-2/IL-15Rβ, CD122) were both expressed in MAIT cells[26,29,30]. However, their relationships to and functions in MAIT cell effector lineages have not been defined. By comparison with Icos−/− mice, we found that ICOS was not detected in CD4+CD8+ double-positive (DP) thymocytes but was detected at low levels in TCRβ+ thymocytes (Supplementary Fig. 1a). ICOS was readily detectable in all three developmental stages of MAIT cells with the lowest and highest expression in stages 1 and 3, respectively (Fig. 1b). Interestingly, CD24−CD44+ stage 3 MAIT cells contain CD122+ICOS^low and CD122−ICOS+ subsets in the thymus (Fig. 1c) and peripheral organs (Supplementary Fig. 1b). As will be shown later in Supplementary Fig. 5e, f, CD122 only expressed on the CD122+ICOS^low subset but not on CD122−ICOS+ subset of MAIT cells or stage 1 and stage 2 MAIT cells. The transcription factors T-bet and RORγt mark MAIT1 and MAIT17, respectively[29]. Stage 1 and 2 MAIT cells did not express these two molecules but stage 3 MAIT cells contain a predominant T-bet−RORγt+ subset (89.933 ± 1.720%, n = 6) and a minor T-bet+RORγt− subset (7.142 ± 1.381%, n = 6) (Fig. 1c). CD122+ICOS^low and CD122−ICOS+ MAIT cells were mostly T-bet+RORγt− and T-bet−RORγt+ in both thymus and peripheral organs (Fig. 1d, e, Supplementary Fig. 1c) and produced predominantly either IFN-γ or IL-17A following PMA and ionomycin stimulation (Fig. 1f) and thereby represented MAIT1 and MAIT17 cells, respectively. Using a multiplex-ELISA, we also detected IFN-γ but not IL-17A production by sorted liver CD122+ICOS^low MAIT1 cells after PMA and ionomycin stimulation. In contrast, and CD122−ICOS+ MAIT17 cells produced IL-17A (Supplementary Fig. 1d). Although IFN-γ was detected in the CD122−ICOS+ cell supernatant, its concentration was much lower than the CD122+ICOS− cells and could be resulted from contamination of MAIT1 cells. Interestingly, both populations produced similar levels of IL-2 and TNF-α. Analysis of single-cell

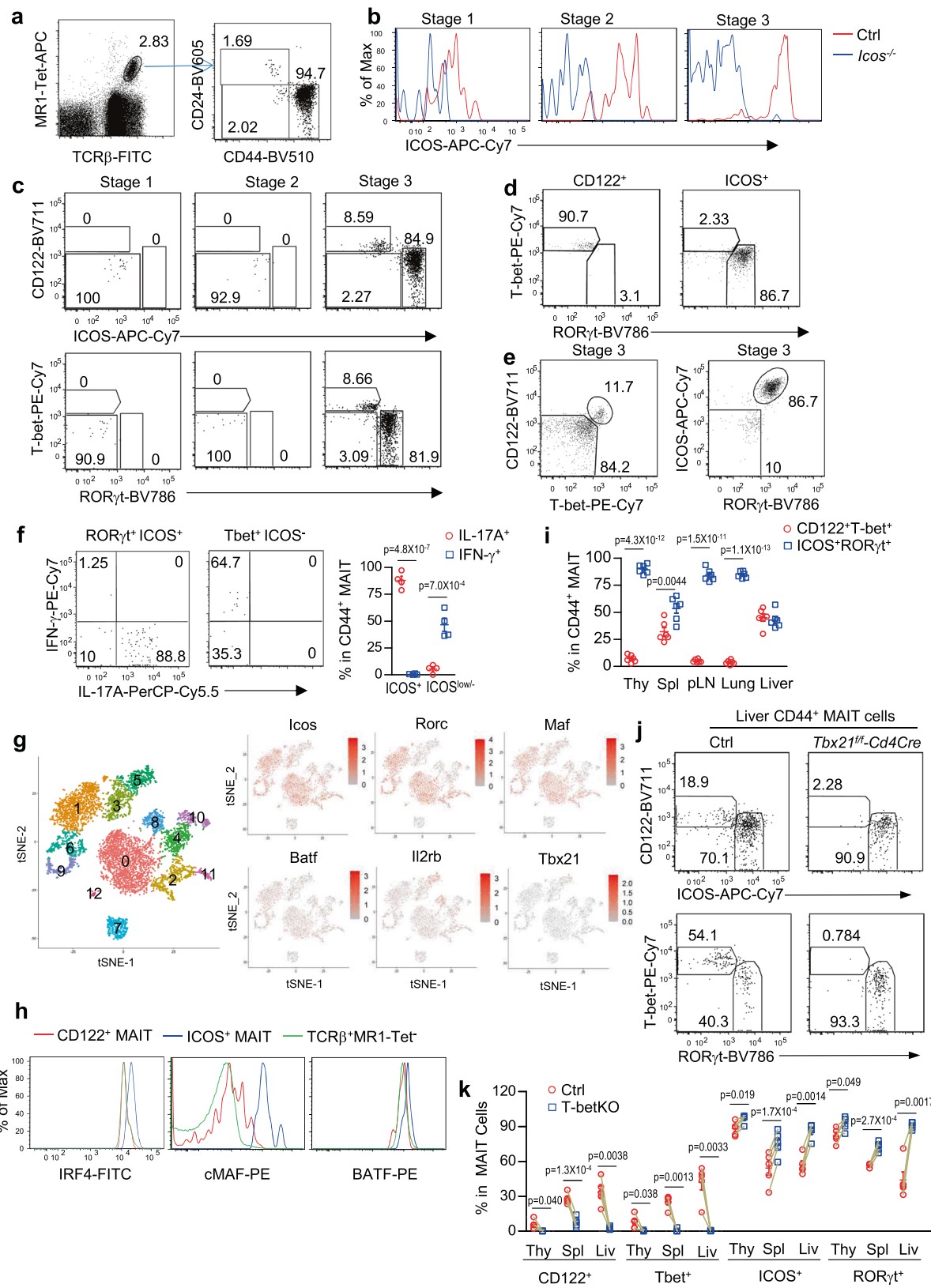

RNA sequencing (scRNAseq) data of thymic MAIT cells generated by Legoux et al.[30], which contained larger numbers of sequenced MAIT cells than other reports[15,28], revealed 13 clusters of MAIT cells (Cluster 0–12, Fig. 1g, Supplementary Fig. 2). Clusters 7 and 10 expressed high levels of *Cd24a* but low levels of *Cd44*, *Icos*, *Rorc* (encoding RORγt), and *Tbx21* (encoding T-bet)

and might represent stage 1 MAIT cells. Clusters 4 and, possibly, 3 were low in both *Cd24a* and *Cd44* and might represent stage 2 MAIT cells. Clusters 5 and 8 expressed higher levels of *Tbx21*, *Il2rb*, and *Ifng* but low levels of *Rorc* and *Icos* might represent MAIT1 cells. The remaining clusters expressed various but higher levels *Rorc* and *Icos* and might represent MAIT17 cells

**Fig. 1 Differential ICOS and CD122 expression in MAIT1 and MAIT17 effector cells. a–i** Thymocytes from 8–10 weeks old WT mice were enriched for MAIT cells with 5-OP-RU-loaded MR1-Tet/magnetic beads and then stained with anti-TCRβ and other lineage antibodies (CD19, PBS-57-loaded CD1d-Tet, B220, CD11b, CD11c, F4/80, Ter119, Gr1, and TCRγδ). **a** Representative FACS plots showing gating strategies for MAIT cells and defining MAIT cell stages. **b** ICOS expression in stages 1–3 MAIT cells. **c** CD122 and ICOS expression, as well as T-bet and RORγt expression in stages 1–3 MAIT cells. **d** T-bet and RORγt expression in CD122+ICOS$^{low}$ and CD122-ICOS+ MAIT cells. **e** CD122 vs T-bet and ICOS vs RORγt staining of stage 3 thymic MAIT cells. **f** Representative FACS plots showing IFN-γ and IL-17A staining in MAIT cells after PMA plus ionomycin stimulation. Scatter plots show percentages of IFN-γ and IL-17A positive cells with the indicated MAIT subsets. **g** tSNE analysis of scRNAseq data from WT thymic MAIT cells. **h** Overlaid histograms show IRF4, cMAF, and BATF levels in thymic MAIT1, MAIT17, and non-MAIT TCRβ+ T cells. **i** Percentages of CD122+T-bet+ MAIT1 and ICOS+RORγt+ MAIT17 cells in different organs. **j, k** Analyses of *Tbx21$^{f/f}$-Cd4Cre* and control mice. **j** Representative FACS plots showing CD122 vs ICOS and T-bet vs RORγt staining in gated liver MAIT cells from WT and *Tbx21$^{f/f}$-Cd4Cre* mice. **k** Percentages of CD122+ICOS$^{low}$, CD122-ICOS+, RORγt-T-bet+, and RORγt+T-bet- cells within control (red circle) and T-bet deficient (blue square) live MAIT cells. Each connection line indicates one pair of control and T-betKO mice examined in one experiment. Data shown were representative of or pooled from at least four experiments. Statistical significance is determined by two-tail unpaired (**f, i**) and pairwise (**k**) Student's *t* test. *P* values of less than 0.05 are shown. All scatter and bar blots in this manuscript are presented as mean ± SEM. Source data for all graphs are provided as a Source Data file.

(Supplementary Fig. 2). Remarkable co-clustering of *Icos* and *Rorc*, as well as *Maf* (encoding cMAF), *Rora, Batf*, and *Fos* expressing cells in clusters 0, 1, 2, 6, 9, 11, and 12 could be observed (Fig. 1g, Supplementary Fig. 3). Interesting, high expressers of *Junb* and *Jund* were found in some but not all of these clusters, suggesting heterogeneity within MAIT17 cells as reported[28,30]. Some of these clusters expressed *Il17a, Il17f, Il22*, and *Il4* mRNA. In contrast, *Txb21* and *Il2rb* expressing cells were mainly limited to *Icos-Rorc-* clusters 5 and 8 that also expressed *Ifng* mRNA (Fig. 1g, Supplementary Fig. 3). Strikingly, *Tgfb1* appeared to be the most frequently expressed cytokine at least at the mRNA level in MAIT cells that spanned both MAIT1 and MAIT17 dominant clusters. Increased expression of cMAF, BATF, and IRF4 proteins in MAIT17 cells were further confirmed by intracellular staining (Fig. 1h). MAIT1 cells expressed similar levels of IRF4 and BATF but slightly increased cMAF proteins compared with conventional αβT cells. However, all these proteins were upregulated in MAIT17 cells (Fig. 1h).

MAIT17 cells were predominant over MAIT1 cells in the thymus, peripheral lymph nodes (pLNs), and lung. However, MAIT1 cell percentages were significantly increased in the spleen and liver, accounting for 32% and 45% total MAIT cells, respectively, in adult mice (Fig. 1i). Together, these observations reveal that CD122 and ICOS can serve as surrogate markers for MAIT1 and MAIT17.

Importantly, CD122+ICOS$^{low}$ MAIT1 cells were dependent on T-bet as they were severely decreased in the thymus, spleen, and liver in T cell specific T-bet deficient (*Tbx21$^{f/f}$-Cd4Cre*, T-betKO) mice. In contrast, CD122-ICOS+ and RORγt+T-bet- MAIT17 cell percentages were concordantly increased and stages 1–3 thymic MAIT cell numbers were not obviously altered in these mice (Fig. 1j, k, Supplementary Fig. 4). These observations further validated CD122 as a marker for MAIT1 cells and revealed a critical but selective role of T-bet for MAIT1 cell development. The notable increases of MAIT17 cells suggested that MAIT1 and MAIT17 are competing fates during MAIT cell effector lineage differentiation.

DAG is a crucial second messenger that relays proximal TCR signal to downstream pathways[59]. We have recently found that downregulation of DAG-mediated signaling and tight control DAG by DGKα/ζ is important for MAIT cell maturation. Overexpression of WT DGKζ (*Dgkz$^{wt}$*) or a gain-of-function mutant of DGKζ (*Dgkz$^{ΔNLS}$*) in developing thymocytes cause modest or severe inhibition of MAIT cell maturation[27,62]. We further examined MAIT1 and MAIT17 cell differentiation in these mice. While overexpression of WT DGKζ did not skew MAIT1 and MAIT17 ratios, the gain-of-function mutant DGKζ$^{ΔNLS}$ caused increased percentages of MAIT1 but decreased percentages of MAIT17 cells without obviously affected ICOS

expression (Supplementary Fig. 1e–g), suggesting that MAIT17 cells are more sensitive to downregulation of DAG-mediated signaling than MAIT1 cells.

**IL-2 and IL-15 signals preferentially promote MAIT1 differentiation.** IL-15 induces upregulation of IFN-γ, TNF-α, granzyme B, and CD69 in human MAIT cells[35,63]. The expression of IL-2/IL-15Rβ on T-bet+ MAIT1 cells prompted us to examine the role of IL-15 signal in murine MAIT cells. In vitro, treatment of WT splenocytes with the IL-15/IL-15RαFc complex induced MAIT cell expansion (Fig. 2a), which resulted mainly from increases of the CD122+ MAIT cells and weakly from increases of the CD122- MAIT cells (Fig. 2b). This was correlated with increased proliferation (Fig. 2c) and improved survival (Fig. 2d). Moreover, T-bet levels were increased in MAIT1 cells after IL-15 treatment (Fig. 2e), suggesting that IL-15R signal upregulated T-bet expression in MAIT1 cells. Injection of IL-15/IL-15RαFc complexes into WT mice induced MAIT cell expansion about two folds in the spleen (Fig. 2f). Within MAIT cells, CD122+ICOS$^{low}$ MAIT1 cell percentages and numbers were both increased; CD122-ICOS+MAIT17 cell percentages were decreased but numbers were not obviously altered in IL-15 treated mice (Fig. 2g, h). Thus, IL-15R signal selectively induces expansion of MAIT1 cells in vitro and in vivo by inducing their proliferation and promoting their survival.

Because IL-2/IL-15Rβ also participates in IL-2 signal and the role of IL-2 for MAIT cells is unknown, we further examined the role of IL-2 signal in MAIT cells. In vitro, IL-2 induced MAIT1 but not MAIT17 cell expansion (Fig. 2i) by enhancing their survival (Fig. 2j) and proliferation (Fig. 2k). In vivo, IL-2/anti-IL-2 antibody complex, which triggers strong signaling[64], also selectively induced MAIT1 but not MAIT17 expansion, leading to relative enrichment of MAIT1 but under-representation of MAIT17 cells (Fig. 2m, n). Additionally, IL-2 stimulation upregulated T-bet levels in MAIT1 cells both in vitro (Fig. 2l) and in vivo without obviously affecting RORγt or ICOS levels in MAIT17 cells (Fig. 2o). Thus, similar to IL-15, IL-2 signal preferentially induced MAIT1 cell expansion and upregulation of T-bet expression in these cells. We further examined whether these two cytokines might exert a synergistic effect on MAIT1 cells. Injection of IL-15/IL-15RαFc complexes into mice alone appeared to induce slightly more MAIT cell expansion than IL-2/anti-IL-2 antibody. However, these two cytokines displayed minimal synergy in inducing MAIT1 cell expansion or T-bet upregulation (Supplementary Fig. 5a, b).

To further determine the requirement of IL-2/IL-15R signal for MAIT cells in vivo, we examined *Il2/15rb$^{f/f}$-Cd2iCre* (IL-2/IL-15βKO) and control *Il2/15rb$^{f/f}$* or *Il2/15rb$^{+/+}$Cd2iCre* mice. Cd2iCre is able to induce recombination between LoxP sites in

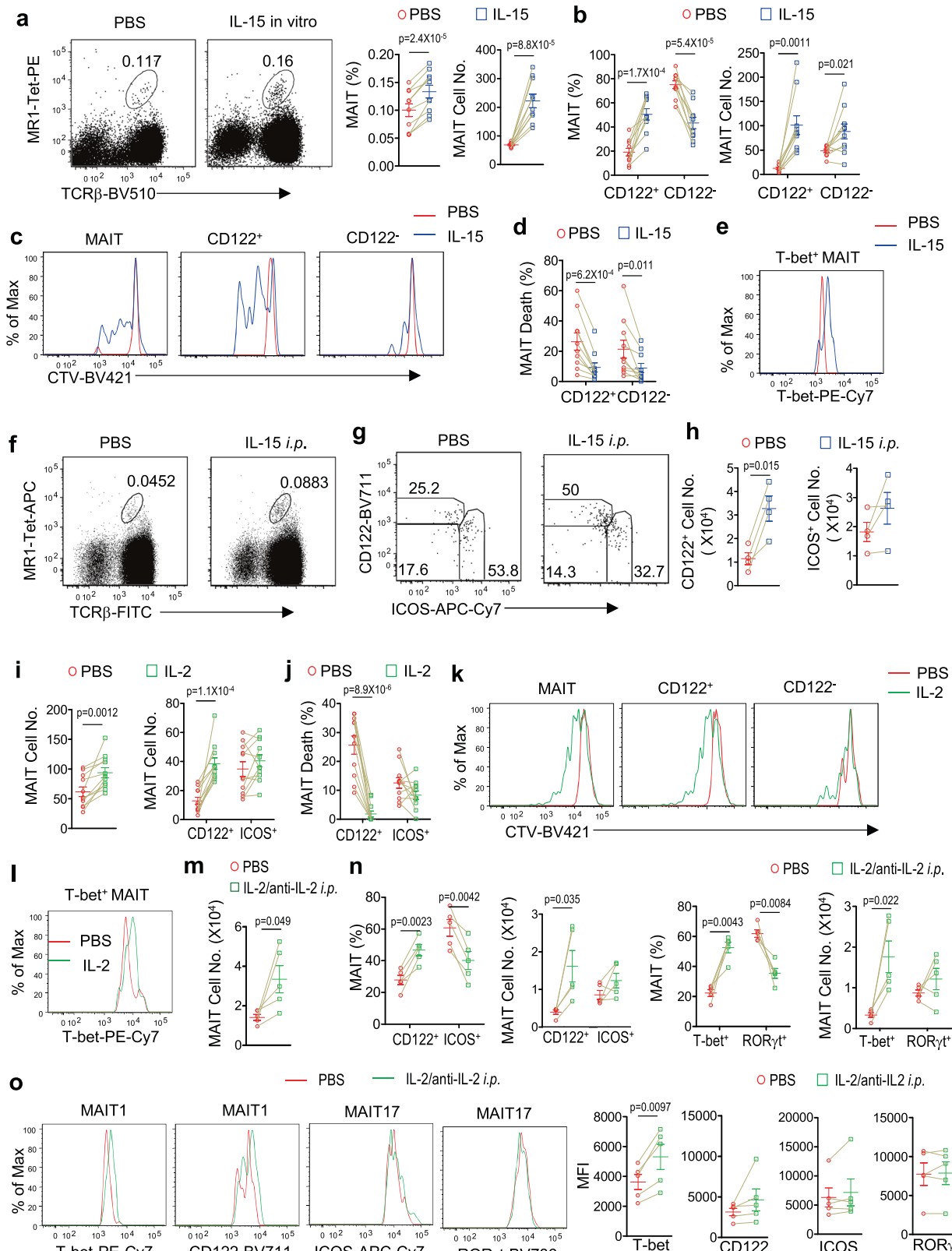

αβT, γδT, and B cells. In IL-2/IL-15RβKO mice, MAIT cell percentages were decreased in the spleen, lung, and liver but not obviously altered in the thymus (Fig. 3a, Supplementary Fig. 5c); their numbers were decreased in all these organs including the thymus (Fig. 3b). Within IL-2/IL-15RβKO thymic MAIT cells, only stage 3 MAIT cell numbers were decreased; stage 1 and stage

2 MAIT cell numbers were not obviously affected (Fig. 3c). The decreases of total thymic MAIT cells were associated with decreases of total thymic cellularity because IL-2/IL-15Rβ deficiency causes severe autoimmune diseases due to impaired regulatory T cells[65]. Within the remaining MAIT cells, the percentages of the T-bet[+] MAIT1 cells were severely decreased

**Fig. 2 IL-15 and IL-2 induce MAIT1 expansion in vitro and in vivo. a–e** CTV-labeled WT splenocytes were cultured with PBS or IL-15/IL-15RαFc in vitro for 72 h. **a** Representative FACS plots showing MAIT cell staining. Scatter plots show MAIT cell percentages and numbers. **b** Scatter plots show CD122+ and CD122− MAIT cell percentages and numbers. **c** Overlaid histogram shows CTV dilution from gated total, CD122+, and CD122− MAIT cells. **d** Death rates of CD122+ and CD122− MAIT cells. **e** Overlaid histogram shows T-bet expression in MAIT1 cells. Data shown are representative of or pooled from ten experiments for (**a–d**) or are representative of eight experiments for (**e**). Each connection line indicates data of one experiment. **f–h** WT mice were i.p. injected with IL-15/IL-15RαFc complexes or PBS and spleens were harvested 72 h after injection. **f** Representative FACS plots showing MAIT cell staining. **g** Representative FACS plots showing CD122 and ICOS staining in MAIT cells. **h** Splenic CD122+ICOS^low and CD122−ICOS+ MAIT cell numbers. Each connected pair represents one experiment. Data are representative of (**f, g**) or pooled from (**h**) four experiments. **i–l** CTV-labeled WT splenocytes were cultured with PBS or IL-2 in vitro for 72 h. **i** Scatter plots show total and ICOS+CD122− or ICOS^lowCD122+ MAIT cell numbers. **j** Death rates of CD122+ and ICOS+ MAIT cells. **k** Overlaid histogram shows CTV dilution from gated total, CD122+, and CD122− MAIT cells. **l** Overlaid histogram of T-bet levels in gated MAIT1 cells. Data shown are pooled from five experiments. **m–o** WT mice were i.p. injected with IL-2/anti-IL-2 antibody complexes or PBS on days 1, 2, and 3 and spleens were harvested on day 5. **m** Total splenic MAIT cell numbers. **n** Splenic CD122+ICOS^low, CD122−ICOS+, T-bet+, and RORγt+ MAIT cell percentages and numbers. **o** Overlaid histograms show T-bet and CD122 expression in MAIT1 cells and RORγt and ICOS expression in MAIT17 cells. Scatter plots show MFI of T-bet, CD122 in MAIT1 cells, and MFI of ICOS, RORγt in MAIT17 cells. Each connected pair represents one experiment. Data are representative of or pooled from five experiments. Statistical significance is determined by two-tailed pairwise Student's *t* test. *P* values of less than 0.05 are shown. Source data for all graphs are provided as a Source Data file.

while the percentages of the RORγt+ MAIT17 cells were not obviously altered with the exception of decreased splenic MAIT 17 cells in IL-2/IL-15RβKO mice (Fig. 3d, e, Supplementary Fig. 5d). However, both T-bet+ MAIT1 and RORγt+ MAIT17 cell numbers were decreased due to severe decreases of total MAIT cells in the mice (Fig. 3f). Because MAIT cells are positively selected by MR1 expressed on DP thymocytes, we further generated and examined irradiation chimeric mice reconstituted with a mixture of CD45.1+ WT and CD45.2+ *Il2/15rb^f/f-Cd2iCre* bone marrow (BM) cells. In the mixed BM chimeric mice, the average CD45.2+ IL-2/IL-15RβKO to CD45.1+ WT ratios of CD4+CD8+ DP thymocytes was 0.76. The ratios for MAIT cells were 0.64, 0.28, and 0.16 in the thymus, spleen, and liver, respectively (Fig. 3g), suggesting reduced presentation of IL-2/IL-15RβKO-derived MAIT cells in the spleen and liver but not in the thymus. Moreover, CD45.2+ IL-2/IL-15RβKO MAIT cells contained severely decreased T-bet+ MAIT1 but increased RORγt+ICOS+ MAIT17 cells compared with CD45.1+ WT MAIT cells in individual recipient mice (Fig. 3h), indicating that IL-2/IL-15Rβ intrinsically promoted MAIT1 cell differentiation. These data also suggested that the decreases of MAIT17 cells in *Il2/15rb^f/f-Cd2iCre* mice shown in Fig. 3e, f was most likely caused by extrinsic factors.

Together, these observations revealed a critical and selective role of IL-15 and IL-2 signals for MAIT1 development and/or homeostasis. However, the possibility that IL-2R and IL-15R signals might weakly promote MAIT17 differentiation/maintenance was not ruled out.

**Intrinsic role of ICOS for MAIT17 differentiation.** A very recent report has found that bacterial-induced lung MAIT cell expression is impaired in ICOS deficient mice[26]. However, the role of ICOS in MAIT cell development was not examined. To determine whether ICOS plays a role in MAIT cell development, we examined *Icos^−/−* mice. ICOS deficiency led to about 70% decreases of percentages and numbers of MAIT cells in the thymus assessed directly (Fig. 4a, b) or after MR1-Tet enrichment (Fig. 4c, d), which was caused by decreased stage 3 but not stages 1 and 2 MAIT cells (Fig. 4e). In fact, stages 1 and 2 MAIT cells were relatively enriched within *Icos^−/−* MAIT cells (Supplementary Fig. 6a, b). Correlated with impaired MAIT cell development, MAIT cell percentages and numbers were also decreased 44 to 65% in the spleen, pLNs, lung, and liver (Fig. 4f).

In *Icos^−/−* thymus, RORγt+ MAIT17 cells were decreased in both percentages and numbers (Fig. 4g, h), correlated with decreased IL-17A- producing MAIT cells in the thymus and pLNs (Fig. 4i). Although T-bet+ MAIT cell percentages were relatively

increased, their numbers were not obviously changed. Thus, ICOS deficiency led to selective impairment of MAIT17 development/maintenance without obviously affecting early MAIT cell development or MAIT1 differentiation.

Because ICOSKO mice are germline deficient, we generated and analyzed irradiation chimeric mice by injection of a mixture of CD45.1+CD45.2+ WT and CD45.1−CD45.2+ ICOSKO BM cells at 1:1 ratio into CD45.1+CD45.2− *TCRJα18^−/−* mice, which lack both iNKT and MAIT cells[66]. Eight weeks after reconstitution, we detected close to 50:50 ratios of WT and ICOSKO derived CD4+CD8+ DP, and CD4+CD8− or CD4−CD8+ SP thymocytes (Fig. 4j), indicating an equal contribution of WT and ICOSKO hematopoietic stem cells to the recipient mice. Within thymic MAIT cells, WT and ICOSKO cells were equally represented at stages 1 and 2 but ICOSKO MAIT cells were obviously under-represented at stage 3, accounting for a quarter of these cells (Fig. 4j, Supplementary Fig. 6c). Within stage 3 MAIT cells, the CD122+ population was close to 1:1 but the CD122− population (predominantly ICOS+ in WT mice) was greatly decreased (Fig. 4k). Additionally, ICOSKO MAIT cells were also under-represented at various degrees in the peripheral organs in the chimeric mice (Fig. 4l). Thus, ICOS deficiency intrinsically impaired MAIT17 but not early MAIT or MAIT1 development.

Together, these data indicate that ICOS intrinsically promotes MAIT17 cell differentiation/maintenance.

**Severe MAIT cell deficiency in the absence of mTOR.** The differential roles of IL-2/IL-15 signals and ICOS costimulatory signal for MAIT1 and MAIT17 cells prompted us to further explore downstream pathways that relay ICOS and IL-2R/IL-15R signals for MAIT cell effector lineage differentiation. We found that both IL-2 and IL-15/IL-15RαFc induced phosphorylation of S6 and Akt S473, indicators of mTORC1 and mTORC2 activation respectively, as well as Akt T308 (indicator of PI3K activation) in MAIT cells. Additionally, both Raptor and Rictor levels were upregulated or weakly upregulated in MAIT cells treated with either IL-15 or IL-2 (Fig. 5a). Moreover, rapamycin could inhibit IL-2 or IL-15/IL-15RαFc induced MAIT cell proliferation (Fig. 5b), suggesting that mTOR played a role downstream of IL-2R and IL-15R in MAIT cells. Both ICOS and TCR signals have been reported to be able to induce mTOR activation in T cells[45,47]. Stimulation of ICOS with an agonist antibody-induced mTORC1 and mTORC2 activation in CD122+ MAIT1 and CD122− mostly MAIT17 cells (Fig. 5c). However, in this in vitro setting, PI3K activation was only induced in CD122− MAIT17 cells. Together, both cytokine and costimulatory signals

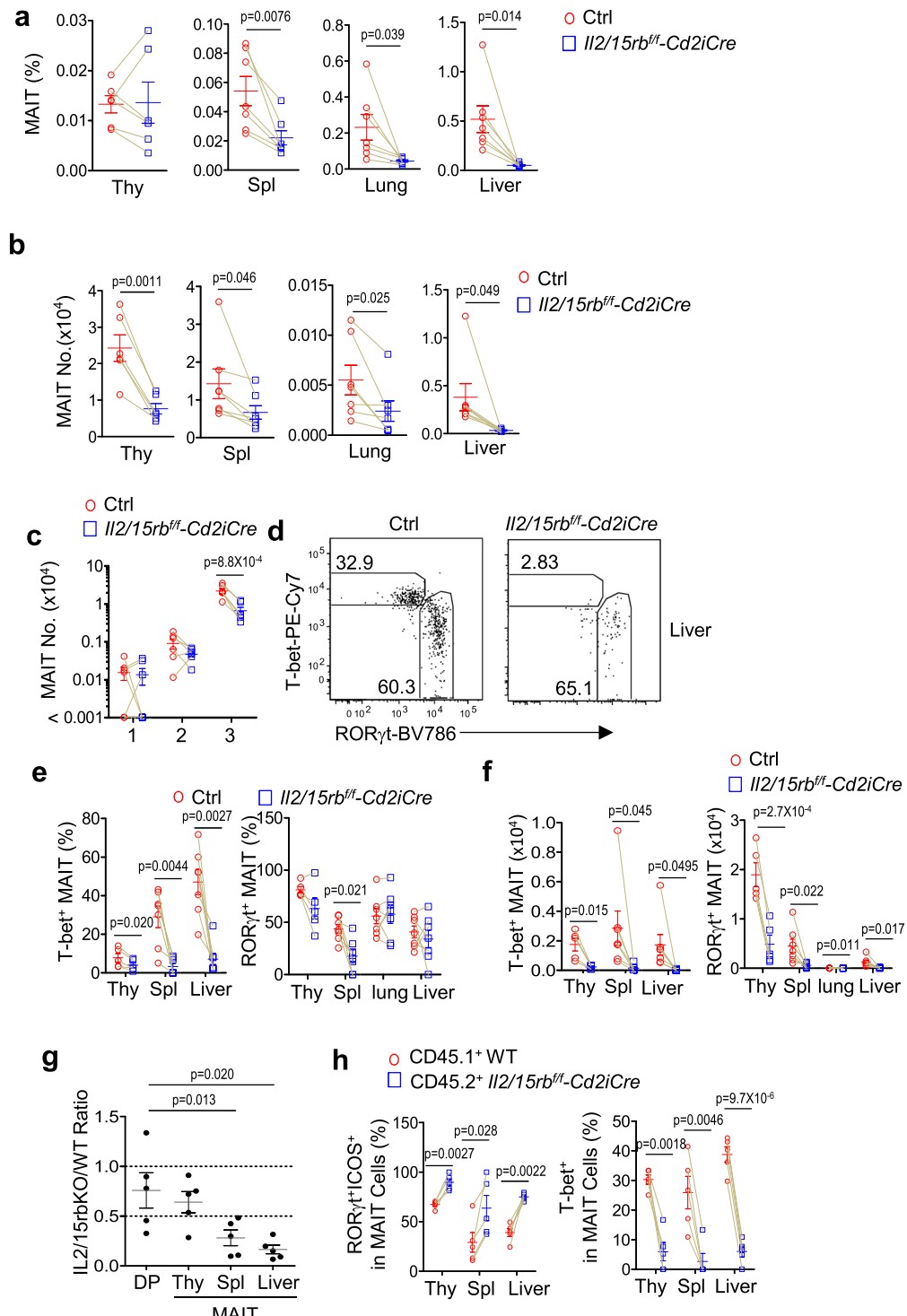

**Fig. 3 Effects of IL-2/IL-15Rβ deficiency on MAIT1 and MAIT17 cells. a–f** Eight–ten weeks old *Il2/15rb^{f/f}-Cd2iCre* and littermate control *Il2/15rb^{f/f}* or WT-*Cd2iCre* mice were analyzed for MAIT cells. **a** MAIT cell percentages in the indicated organs. **b** MAIT cell numbers in the indicated organs. **c** Stages 1–3 thymic MAIT cell numbers. **d** Representative FACS plots of T-bet and RORγt staining in liver CD24⁻CD44⁺ TCRβ⁺MR1-Tet⁺ MAIT cells. **e, f** T-bet⁺ MAIT1 and RORγt⁺ MAIT17 percentages (**e**) and numbers (**f**) in the indicated organs. Each connecting line represents one pair of control and IL-2/IL-15RβKO mice examined in one experiment. Data shown are pooled from or are representative of at least five experiments. **g, h** Analyses of lethally irradiated CD45.1⁺CD45.2⁺ recipient mice 2 months after being reconstituted with a mixture of BM cells from CD45.1⁺ WT and CD45.2⁺ *Il2/15rb^{f/f}-Cd2iCre* mice at 1:1 ratio. **g** CD45.2⁺ IL-2/IL-15RβKO/CD45.1⁺ WT ratios of DP thymocytes and MAIT cells in the thymus, spleen, and liver. Each circle represents one recipient mouse. Bars represent mean ± SEM. **h** Percentages of MAIT1 and MAIT17 cells within CD45.2⁺ IL-2/IL-15RβKO and CD45.1⁺ WT MAIT cells. Each connecting line represents WT and IL-2/IL-15RβKO MAIT cells in one recipient mouse. Data shown are pooled from five experiments. Statistical significance is determined by two-tailed pairwise Student's *t* test. *P* values of less than 0.05 are shown. Source data for all graphs are provided as a Source Data file.

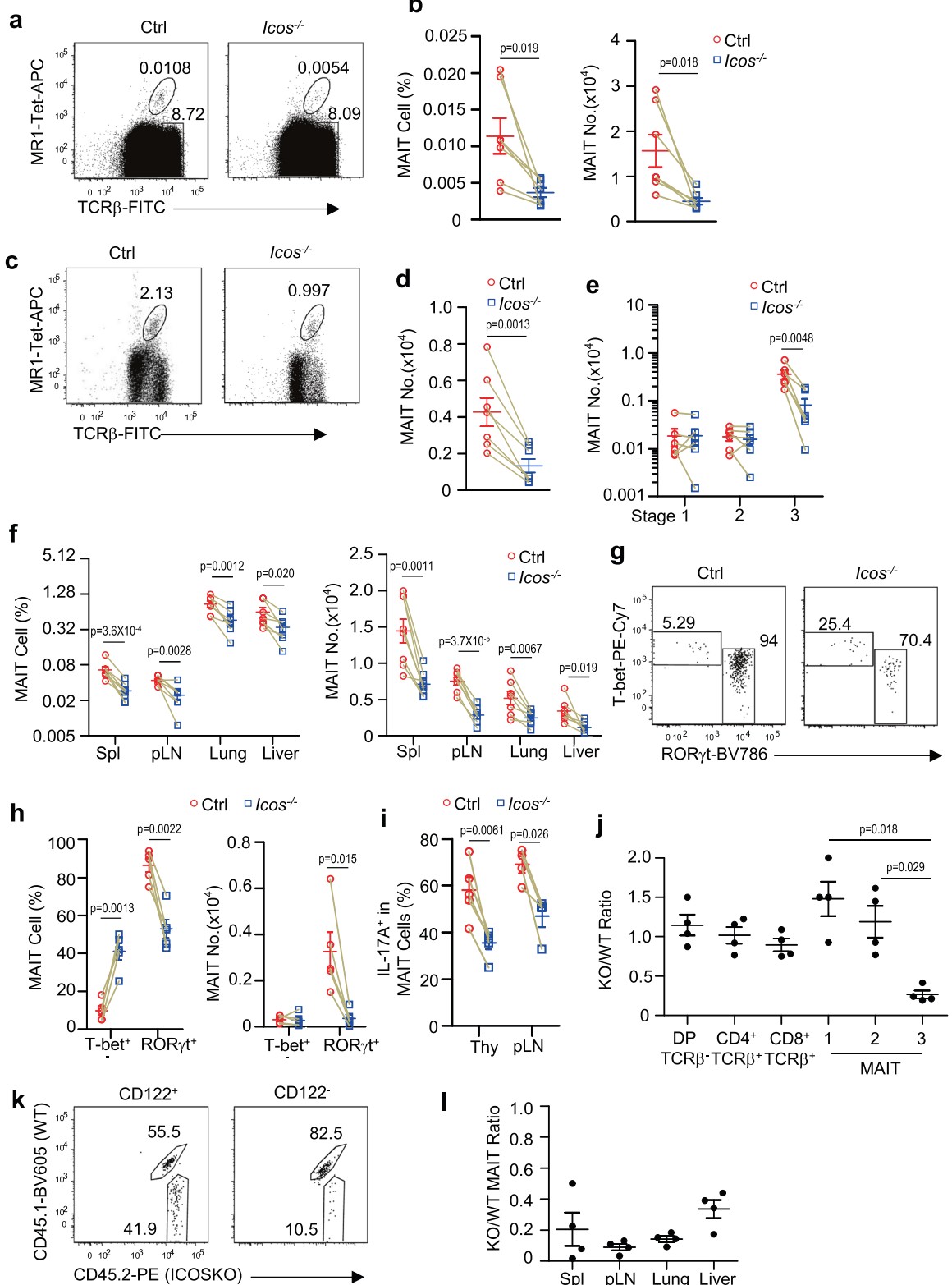

could induce mTORC1 and mTORC2, as well as PI3K activation in MAIT cells.

To determine the role of mTOR in MAIT cells, we analyzed *mTOR^{f/f}-Cd4Cre* (mTORKO) mice. MAIT cells were drastically decreased in the thymus (Fig. 5d–g) and in other peripheral organs (Fig. 5h, Supplementary Fig. 7). TCRVα19-Jα33

recombination in *mTOR^{f/f}-Cd4Cre* DP thymocytes was not decreased (Fig. 5i), suggesting that such developmental defect was not caused by impaired recombination of the MAIT specific TCR. Within thymic mTORKO MAIT cells, stage 1 and stage 2 MAIT cell percentages were increased but numbers were similar to controls, while stage 3 MAIT cells were virtually absent

**Fig. 4 ICOS intrinsically promotes MAIT17 development. a–i** Thymocytes and single-cell suspension of the indicated organs from 8–10 weeks old $Icos^{-/-}$ and control mice were stained for MAIT cells similarly to Fig. 1 directly (**a**, **b**, **f**) or after enrichment with MR1-Tet (**c–e**, **g–i**). **a**, **c** Dot plots show TCRβ and MR1-Tet staining in live gated Lin⁻ cells before (**a**) and after (**c**) enrichment. **b**, **d** Scatter plots show MAIT cell percentages and numbers. **e** Stages 1–3 MAIT cell numbers in the thymus. **f** MAIT cell percentage and numbers in the indicated organs. **g** T-bet and RORγt expression in thymic MAIT cells. **h** Percentages and numbers of thymic MAIT1 and MAIT17 cells. **i** Percentages of IL-17A⁺ thymic and pLN MAIT cells after ex vivo PMA plus ionomycin stimulation for 4 h. Each circle and square represents one WT and ICOSKO mice, respectively. Data shown are representative of or pooled from four to eight experiments. **j–l** Analyses of irradiation chimeric CD45.1⁺ $TCRJα18^{-/-}$ mice reconstituted with CD45.1⁺CD45.2⁺ WT and CD45.2⁺ $Icos^{-/-}$ BM cells. **j** ICOSKO to WT ratios of the indicated populations of cells in the thymus from individual recipient mice. **k** Representative dot plots show CD45.1 and CD45.2 expression in CD122⁺ and CD122⁻ stage 3 MAIT cells. **l** ICOSKO to WT ratios of MAIT cells in the indicated organs from individual recipient mice. Each circle represents one recipient mouse. Data shown are representative of or pooled from four experiments. Statistical significance is determined by two-tailed pairwise Student's $t$ test. $P$ values of less than 0.05 are shown. Source data for all graphs are provided as a Source Data file.

(Fig. 5j, k), indicating that mTOR was dispensable for early MAIT cell development but was pivotal for stage 3 functional maturation of MAIT cells.

Expression of PLZF, which is critical for MAIT cell functional maturation[29], was similar in WT and mTORKO stages 1 and 2 MAIT cells (Fig. 5l), suggesting that defective functional maturation of mTOR deficient MAIT cells was not caused by decreased PLZF expression. In irradiation BM chimeric mice (CD45.1⁺CD45.2⁺) reconstituted with a mixture of CD45.1⁺ WT and CD45.2⁺ mTORKO BM cells, the average ratios of mTORKO to WT derived cells were about 0.86:1 for DN thymocytes (suggesting close to equal reconstitution) and 0.40:1 for DP thymocytes (suggesting a moderate role of mTOR for DP thymocyte development/expansion). The ratios were 0.30 and 0.45:1 for stages 1 and 2 MAIT cells, respectively, which were close to the ratios of DP thymocytes. However, no stage 3 mTORKO MAIT cells were detected (Fig. 5m).

Together, these observations demonstrate that mTOR plays an intrinsic and pivotal role for MAIT cell development into stage 3 to gain effector function but is dispensable for stages 1 and 2 MAIT cell development and that such function of mTOR is not mediated by controlling PLZF expression. It has been reported that mTORC1 promotes PLZF nuclear localization in iNKT cells[52], the extremely low numbers of MAIT cells in mTOR deficient mice prevented us from clear determination whether PLZF nuclear localization was altered in mTOR deficient MAIT cells.

**Critical role of mTORC1 for MAIT cell, especially MAIT1 cell, development.** mTOR signals via mTORC1 and mTORC2. To further illustrate the roles of these complexes in MAIT cells, we analyzed T cell-specific Raptor-deficient (mTORC1KO, $Raptor^{f/f}$-$Cd4Cre$) mice. mTORC1KO mice displayed similar but slightly less severe phenotypes than mTORKO mice. They had severe decreases of MAIT cell percentages and numbers in the thymus (Fig. 6a–d) and peripheral organs (Fig. 6e, Supplementary Fig. 8), relative increases of stages 1 and 2 but decreases of stage 3 MAIT cell ratios (Fig. 6f, g), normal numbers of stage 1 and stage 2 but drastic decreases of stage 3 MAIT cells in the thymus (Fig. 6f, g). Thus, mTORC1 is not required for early MAIT cell development but is critical for MAIT cell functional maturation. Within the remaining stage 3 mTORC1KO MAIT cells, the CD122⁺T-bet⁺ MAIT1 cells were virtually undetectable, resulting in about 93.8% decreases for percentages and 99.7% decreases for numbers (Fig. 6h, i). However, ICOS⁺RORγt⁺ MAIT17 cell percentages were not decreased or even increased, although their numbers were decreased. Within the remaining MAIT17 cells, RORγt and IRF4 levels were not obviously different from WT controls (Fig. 6j). Because there were virtually no MAIT1 cells in mTORC1KO mice, we examined the effects of rapamycin on IL-2- and IL-15-induced T-bet upregulation in WT MAIT1 cells and found that rapamycin potently inhibited T-bet upregulation after

IL-2 and IL-15 treatment (Fig. 6k). Thus, mTORC1 plays a more critical role for MAIT1 cell development than for MAIT17 cells, although it is important for the accumulation of MAIT17 cells. PLZF levels were similar in stages 1 and 3 MAIT cells and increased in stage 2 MAIT cells from mTORC1KO thymus (Fig. 6l), suggesting that impaired MAIT cell effector lineage differentiation was not caused reduced PLZF protein expression. The increase of PLZF levels in mTORC1KO stage 2 MAIT cells could be resulted from developmental blockade to stage 3, leading to the accumulation of this protein inside the cells. Together, these observations indicate that mTORC1 contributes greatly to the expansion of stage 3 MAIT cells and that MAIT1 lineage is more stringently dependent on mTORC1 than MAIT17 lineage.

**Critical role of mTORC2 signaling in MAIT17 cell development.** To evaluate the role of mTORC2 in MAIT cell development, we examined $Rictor^{f/f}$-$Cd4Cre$ (mTORC2KO) and control mice. In mTORC2KO mice, MAIT cells were drastically decreased in both percentages and numbers in the thymus (Fig. 7a, b) and peripheral organs (Fig. 7c, Supplementary Fig. 9a, b), indicating a critical role of mTORC2 for MAIT cell development and/or maintenance. In CD45.1⁺CD45.2⁺ irradiated recipient mice reconstituted with equal numbers of CD45.1⁺ WT and CD45.2⁺ $Rictor^{f/f}$-$Cd4Cre$ BM cells, $Rictor^{f/f}$-$Cd4Cre$-derived CD24⁺ MAIT cells were decreased and CD24⁻ MAIT cells were drastically decreased compared with DP thymocytes (Fig. 7d), indicating that mTORC2 was intrinsically required for MAIT cell, especially for late-stage MAIT cell development. The remaining MAIT cells in $Rictor^{f/f}$-$Cd4Cre$ thymus, contained relatively increased stages 1 and 2 but decreased stage 3 MAIT cells. Due to severe decreases of total MAIT cells in mTORC2KO thymus, their stages 1, 2, and 3 MAIT cell numbers were decreased 49.2, 57.6, and 95.8%, respectively (Fig. 7e, f). These data were consistent with data shown in Fig. 7d that mTORC2 deficiency caused weak and strong developmental defects during early and late MAIT cell development, respectively. Thus, mTORC2 exerted a weak role for early MAIT cell development but played a critical role for stage 3 MAIT cell maturation/homeostasis. Because mTOR deficient mice did not show obvious early MAIT cell developmental defect, the presence of mTORC1 activity or potentially dysregulated mTORC1 activity in the absence of mTORC2 might contribute to the partial developmental defects of stages 1 and 2 MAIT cells in mTORC2KO mice.

Stage 3 MAIT cells in mTORC2KO thymus contained increased percentages of CD122⁺T-bet⁺ MAIT1 cells but drastically decreased ICOS⁺RORγt⁺ MAIT17 cells (Fig. 7g–i), accompanying increased percentages of IFN-γ- but decreased IL-17A-producing MAIT cells (Fig. 7j). Due to severe reduction of total MAIT cells, the total numbers of CD122⁺ and T-bet⁺ MAIT1 cells were actually decreased 52.0 and 87.6%, respectively (Fig. 7h, i). The magnitude of reduction of MAIT1 cells was in line with or partially in line with the decreases of stage 1/2

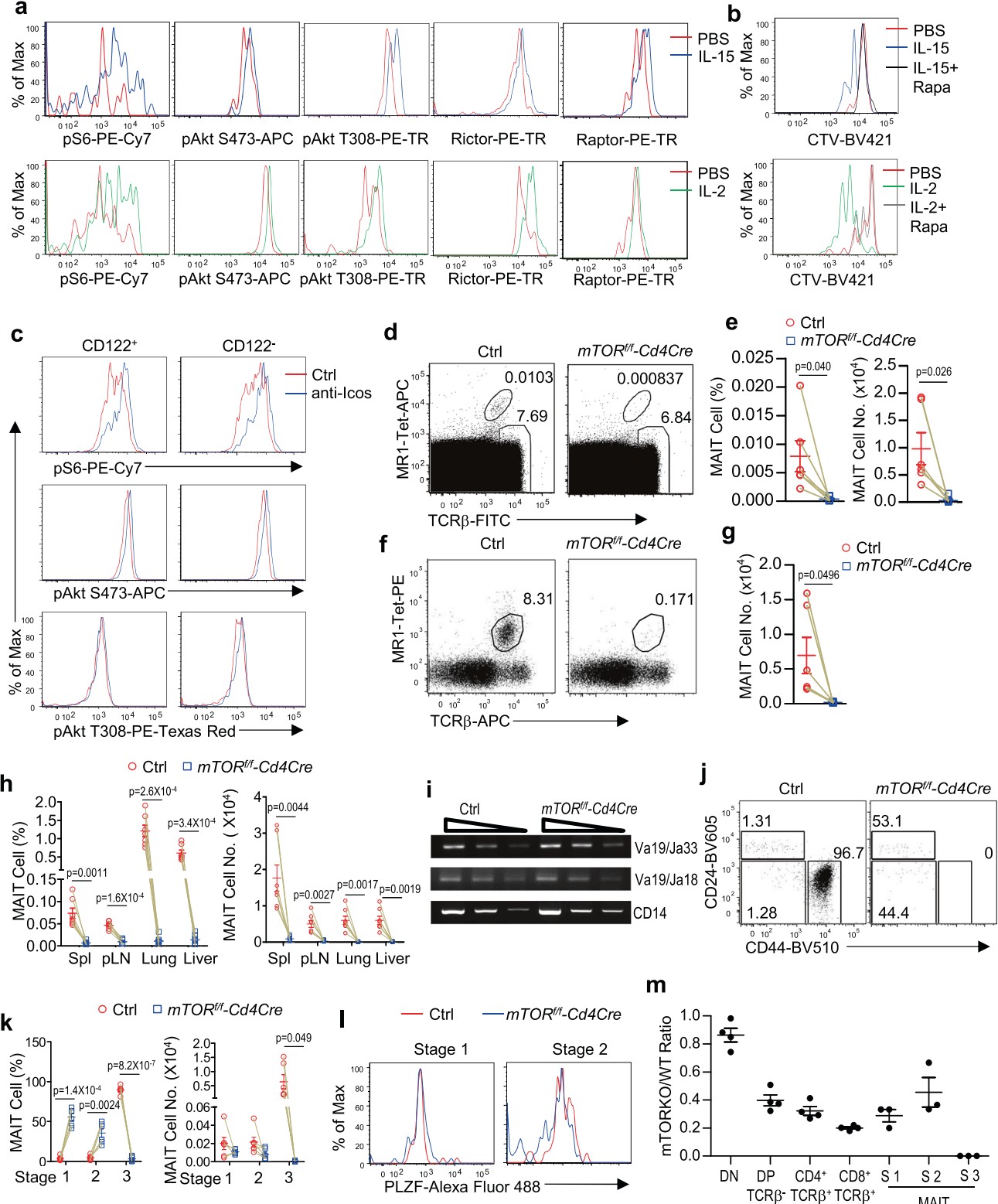

MAIT cells and was much less than the 99.8% decrease of MAIT17 cells. Within mTORC2KO MAIT1 cells, T-bet and CD122 levels were similar to WT controls (Fig. 7k, l), suggesting mTORC2 did not directly control T-bet or CD122 expression. mTORC2KO MAIT17 cells expressed similar levels of RORγt and IRF4, suggesting that mTORC2 might not directly promote the expression of these transcription factors to augment MAIT17 differentiation. However, mTORC2KO MAIT17 cells had

obviously decreased cMAF levels (Fig. 7k, l). As cMAF plays critical roles for Th17, γδT17, and iNKT17 lineage differentiation[67–69], mTORC2 might promote cMAF expression to direct MAIT17 differentiation. Together, mTORC2 contributes to early MAIT cell development and MAIT1 development/ maintenance but is pivotal for MAIT17 cell differentiation/ maintenance likely via promoting cMAF expression.

**Fig. 5 Crucial role of mTOR for MAIT cell functional maturation. a–c** CTV-labeled or unlabeled WT splenocytes were cultured in vitro in the presence or absence of IL-15/IL-15RαFc or IL-2 for 72 h (**a**, **b**) or with an anti-ICOS antibody for 30 min (**c**). **a** Overlaid histograms show phosphor-S6, phosphor-Akt S473, phosphor-Akt T308, and intracellular Raptor and Rictor staining in live gated Lin⁻ WT MAIT cells. **b** Overlaid histogram shows CTV dilution from live gated Lin⁻ MAIT cells in the presence or absence of rapamycin. **c** Overlaid histograms show phosphor-S6, phosphor-Akt S473, and phosphor-Akt T308 staining in live gated Lin⁻ WT MAIT cells with an anti-ICOS antibody for 30 min. Data shown are representative of three to nine experiments for (**a**), six experiments for (**b**), and five experiments for (**c**). **d–l** Thymocytes and single-cell suspension of the indicated organs from 8–10 weeks *mTOR^f/f-Cd4Cre* and control *mTOR^f/f* or *WT-Cd4Cre* mice were stained for MAIT cells similar to Fig. 1 directly (**d**, **e**, **h**) or after enrichment with 5-OP-RU-loaded MR1-Tet (**f**, **g**, **j–l**). **d** FACS plots show TCRβ and MR1-Tet staining in live gated Lin⁻ thymocytes without enrichment. **e** Percentages and numbers of unenriched thymic MAIT cells. **f** FACS plots show TCRβ and MR1-Tet staining in live gated Lin⁻ thymocytes after enrichment. **g** Thymic MAIT cell numbers after enrichment. **h** MAIT cell percentages and numbers in peripheral organs. **i** Semi-quantitative PCR detection of Vα19-Jα33 and Vα19-Jα18 recombination in CD4⁺CD8⁺TCRβ⁻ thymocytes. Cd14 serves as PCR control. Vα19-Jα33, Vα19-Jα18, and Cd14 PCR products are about 120, 80, and 471 base pairs, respectively. The gel image and DNA markers can be found in the Source Data file. **j** Representative FACS plots showing CD44 and CD24 straining in enriched thymic MAIT cells. **k** Scatter plots show stage 1–3 thymic MAIT cell percentages and numbers after enrichment. **l** Overlaid histogram showing PLZF expression in thymic stages 1 and 2 MAIT cells. Data in **d–h**, **j**, and **k** are representative of or pooled from six to seven experiments. Data in **i** and **l** are representative of three experiments. Each circle and square represents one WT control and mTORKO mice, respectively. Connection line represents one pair of WT control and mTORKO mice analyzed in one experiment. **m** mTORKO/WT ratios of the indicated thymocyte populations from individual CD45.1⁺CD45.2⁺ recipient mice two months after irradiation and reconstitution with a mixture of CD45.1⁺ WT and CD45.2⁺ *mTOR^f/f-Cd4Cre* BM cells. Each circle represents one recipient mouse. Bars represent mean ± SEM. Data shown are pooled from three to four experiments. Statistical significance is determined by two-tailed pairwise Student's *t* test. *P* values of less than 0.05 are shown. Source data for all graphs are provided as a Source Data file.

Within mTORC2KO MAIT cells, PLZF expression was not altered in stage 1, increased in stage 2, but decreased in stage 3 (Fig. 7m). The increase of PLZF expression at stage 2 suggested that mTORC2 was not required for PLZF expression. The decreases of PLZF expression in stage 3 mTORC2KO MAIT cells were likely caused by the severe decreases of MAIT17 cells, which is the predominant population of MAIT cells expressing higher levels of PLZF than MAIT1 cells. Within the few remaining mTORC2KO RORγt⁺ stage 3 MAIT cells, PLZF levels were actually higher than WT control (Fig. 7m), suggesting that mTORC2 may not enhance MAIT17 development by promoting PLZF expression. Interestingly, ICOS expression was decreased in both stage 2 and stage 3 MAIT cells (Fig. 7n), suggesting that mTORC2 was involved in the upregulation of ICOS expression in MAIT cells to promote MAIT17 differentiation. It has also been reported that ICOS induces cMAF expression to promote Th17 differentiation and IL-17 production[69]. Decreased ICOS expression in mTORC2 MAIT cells could similarly lead to reduced cMAF expression and subsequent MAIT17 differentiation.

**IL-1β and IL-23 potently induce MAIT cell expansion via mTOR.** Although both ICOS and mTORC2 promoted MAIT17 differentiation, mTORC2 deficiency resulted in much more severe MAIT17 defect than ICOS deficient mice, suggesting that mTOR may integrate additional signals to promote MAIT17 differentiation. It has been reported that IL-1β and IL-23 promote Th17 differentiation via mTOR[70]. Interestingly, IL-1β and IL-23 stimulation of WT splenocytes induced considerable MAIT cell expansion (Fig. 8a, b) that was correlated with increased survival (Fig. 8c) and proliferation (Fig. 8d). Such treatment also induced mTORC1 and mTORC2 activation indicated by S6 and Akt S473 phosphorylation in MAIT cells (Fig. 8e). Moreover, IL-1β or IL-23 alone was able to promote MAIT cell expansion by enhancing survival and proliferation (Fig. 8f–h). IL-1β displayed stronger effects than IL-23 on MAIT cell expansion that was mostly resulted from more vigorous proliferation (Fig. 8g) and these two cytokines appeared to synergize with each other to enhance MAIT cell expansion. Importantly, the effects of IL-1β and IL-23 on MAIT cell expansion were dependent on mTOR as such expansion was severely inhibited by rapamycin (Fig. 8f, g).

To further determine whether IL-1β and IL-23 regulate MAIT cells in vivo, we injected IL-1β and IL-23 individually or together into WT mice on day 1 and day 2 and examined the mice on day 4. IL-1β and IL-23 could weakly induce MAIT cell expansion in the lung and liver by themselves (Fig. 8i, j), which correlated with enhanced proliferation (Fig. 8k). Such effects were much stronger in mice coinjected with both IL-1β and IL-23. Consistent with in vitro observations, IL-1β and IL-23 induced MAIT cell expansion was substantially inhibited when coinjected with rapamycin (Fig. 8l). Thus, IL-1β and IL-23 promoted MAIT cell expansion in vivo weakly by themselves but strongly when administrated together and such expansion was dependent on mTOR activity.

**IL-1β and IL-23 influence MAIT effector lineages in vitro and in vivo.** In vitro, IL-23, IL-1β, or IL-23 plus IL-1β treatment led to decreases of the relative ratios of RORγt⁻T-bet⁺ MAIT1 cells. In contrast, RORγt⁺T-bet⁻ MAIT17 ratios were only increased after IL-23 but not IL-1β or IL-1β plus IL-23 treatment (Fig. 9a, b). IL-23 by itself had no obvious effect on MAIT1 cell numbers but could weakly induce expansion of MAIT17 cells. In contrast, IL-1β could induce both MAIT1 and MAIT17 cell expansion and such effects appeared stronger on MAIT17 cells than on MAIT1 cells (Fig. 9c). Interestingly, a T-bet⁺RORγt⁺ population of MAIT cells was induced by IL-1β or IL-1β plus IL-23 but not by IL-23 alone (Fig. 9a–c). IL-1β but not IL-23 also increased RORγt levels in MAIT17 cells (Fig. 9d). Thus, in these in vitro experiments, IL-1β promoted MAIT1 and, more strongly, MAIT17 cell, as well as the T-bet⁺RORγt⁺ cell expansion. IL-23 selectively promoted MAIT17 expansion and can synergize with IL-1β during MAIT cell expansion.

In the liver of IL-1β, IL-23, or IL-1β plus IL-23 injected WT mice, the relative percentages of T-bet⁺RORγt⁻ MAIT1 cells were slightly decreased in IL-1β and most severely in IL-1β plus IL-23 injected mice. However, their numbers were not obviously altered (Fig. 9e, f). The percentages of T-bet⁻RORγt⁺ MAIT17 cells were not increased but their numbers displayed increased trends in IL-1β plus IL-23 treated mice although not statistically significant (*p* = 0.093). Interesting, the Tbet⁺RORγt⁺ MAIT cells that were very rare in untreated mice were drastically increased in both percentages and numbers in mice injected with either IL-1β, IL-23, or more strikingly, both IL-1β and IL-23 (Fig. 9e, f). This population of MAIT cells was also expanded considerably in the lung in mice co-injected with IL-1β and IL-23 (Fig. 9g). These T-bet⁺RORγt⁺ MAIT cells expressed similar low levels of CD122 and similarly high levels of ICOS as T-bet⁻RORγt⁺ MAIT17 cells and were different

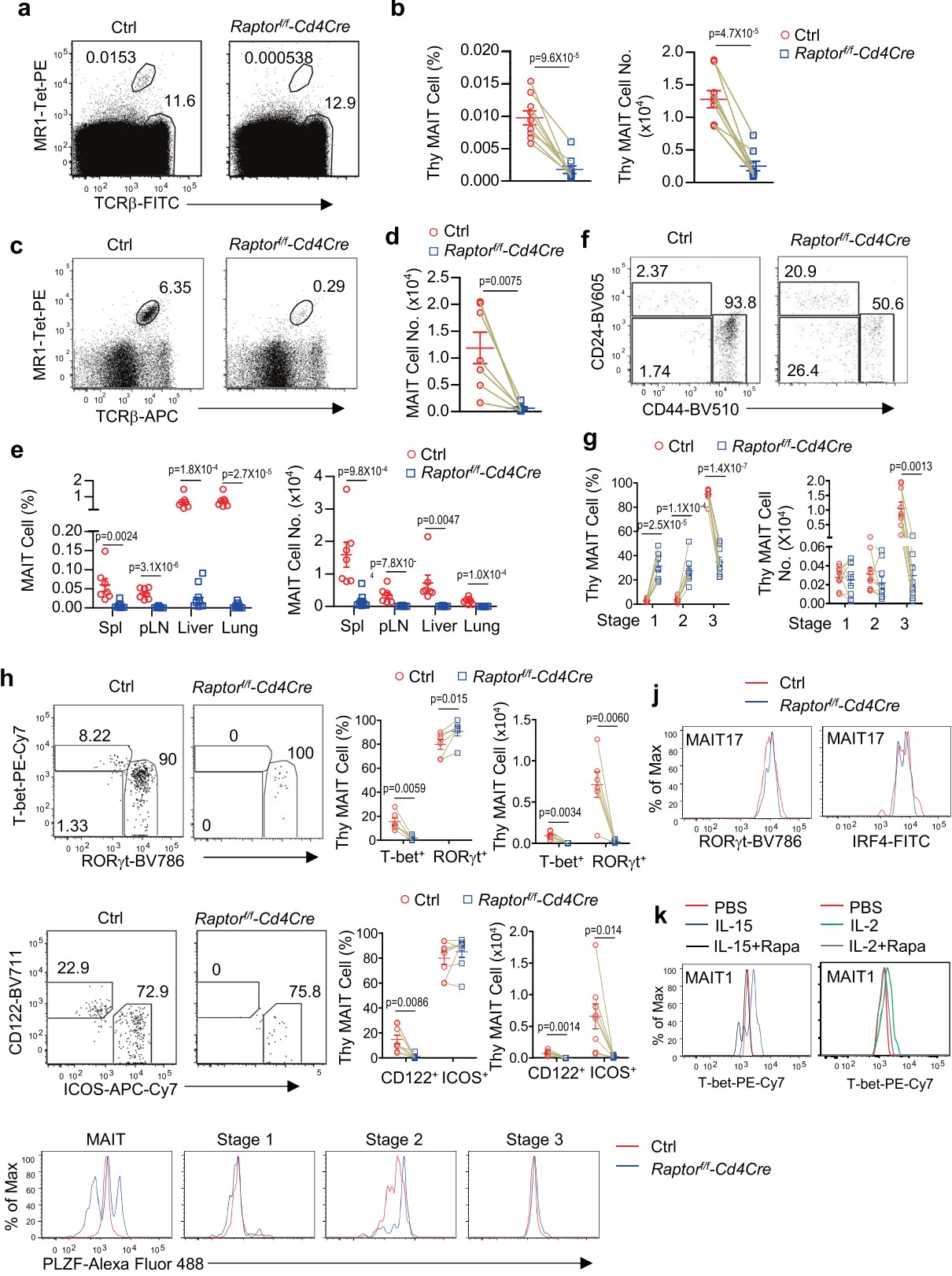

from the T-bet⁺RORγt⁻ MAIT1 cells (Fig. 9h), suggesting that these T-bet⁺RORγt⁺ cells were most likely derived from T-bet⁻RORγt⁺ MAIT17 cells. Moreover, following in vitro stimulation of sorted CD24⁻CD44⁺ICOS⁻CD122⁺ MAIT1 and CD24⁻CD44⁺ICOS⁺ CD122⁻ MAIT17 cells with IL-1β and IL-23, ICOS⁺CD122⁻ MAIT17 cells but not ICOS⁻CD122⁺ MAIT1 cells expanded

(Fig. 9i left panel) and a portion of ICOS⁺CD122⁻ MAIT17 cells but not ICOS⁻CD122⁺ MAIT1 cells turned into T-bet⁺RORγt⁺ cells (Fig. 9i right panel). These data suggested that MAIT17 but not MAIT1 cells responded to in vitro IL-1β and IL-23 for expansion and that T-bet⁺RORγt⁺ MAIT cells were derived from ICOS⁺ CD122⁻RORγt⁺ MAIT17 cells.

**Fig. 6 Critical role of mTORC1 for MAIT cell effector lineage differentiation.** Thymocytes (**a–d**, **f–j**, and **l**) and single-cell suspension of the indicated peripheral organs (**e**) from 8–10 weeks old Raptor$^{f/f}$-Cd4Cre and control Raptor$^{f/f}$ or WT-Cd4Cre mice were stained for MAIT cells similar to Fig. 1 directly (**a**, **b**) or after enrichment with 5-OP-RU-loaded MR1-Tet (**c**, **d**, **f–j**, and **l**). **a** FACS plots show TCRβ and MR1-Tet staining in live gated Lin$^-$ thymocytes without enrichment. **b** Percentages and numbers of unenriched thymic MAIT cells. **c** FACS plots show TCRβ and MR1-Tet staining in live gated Lin$^-$ thymocytes after enrichment. **d** Thymic MAIT cell numbers after enrichment. **e** MAIT cell percentages and numbers in peripheral organs. **f** Representative FACS plots showing CD44 and CD24 staining in enriched thymic MAIT cells. **g** Scatter plots show stages 1–3 thymic MAIT cell percentages and numbers after enrichment. **h** T-bet and RORγt staining in live gated Lin$^-$ CD44$^+$ thymic MAIT cells after enrichment. Scatter plots show T-bet$^+$ and RORγt$^+$ MAIT cell percentages and numbers after enrichment. **i** CD122 and ICOS staining in live gated Lin$^-$ CD44$^+$ thymic MAIT cells. Scatter plots show CD122$^+$ and ICOS$^+$ MAIT cell percentages and numbers after enrichment. **j** Overlaid histograms show expression of indicated molecules in ICOS$^+$CD122$^-$ MAIT17 cells. **k** Overlaid histograms show T-bet levels in MAIT1 cells after IL-15 or IL-2 stimulation in the presence or absence of rapamycin for 3 days. **l** Overlaid histograms show PLZF levels in different stages of MAIT cells. Each circle and square represents one control and mTORC1KO mice, respectively. Connection line represents one pair of control and mTORC1KO mice analyzed in one experiment. Data shown are representative or pooled from four to nine experiments. Statistical significance is determined by two-tailed pairwise Student's t test (**b**, **d**, **g–i**) or two-tailed unpaired Student's t test (**e**). P values of less than 0.05 are shown. Source data for all graphs are provided as a Source Data file.

We further found WT MAIT cells express IL-1 receptor IL-1R1 with low levels at stage 1 and intermediate levels at stage 3. Stage 2 MAIT cells appeared to contain populations that expressed either intermediate or high levels of IL-1R1 (Fig. 9j, top panels). In addition, both MAIT1 and MAIT17 cells expressed similar levels of IL-1R1 (Fig. 9j, bottom panels). Thus, MAIT cells are developmentally equipped with the capacity to sense proinflammatory cytokines for expansion in an mTOR-dependent manner. The ability of IL-1β to induce MAIT1 cell expansion in vivo but not in vitro suggests that additional factors are involved in MAIT1 cell response to this cytokine in vivo.

## Discussion

Our data reveals that ICOS and CD122 (IL-2/IL-15Rβ) are developmentally regulated in MAIT cells and can be used to identify MAIT1 and MAIT17 cells. ICOS has been considered inducibly expressed in activated T cells. Using ICOS deficient mice as control, we have revealed that ICOS is not expressed in DP thymocytes but expressed at low levels in cαβT cells in the thymus and peripheral organs in the steady state. ICOS is also expressed at low levels in stage 1 MAIT cells but progressively upregulated as they mature. Different from ICOS, CD122 is expressed only in a small portion of thymic stage 3 MAIT cells. Most importantly, CD122 and ICOS can serve as lineage-specific markers for MAIT1 and MAIT17 cells, respectively. Stage 3 MAIT cells contain a predominate ICOS$^+$CD122$^-$ population that is also T-bet$^-$RORγt$^+$ and IL-17A-producing MAIT17 cells and a minor ICOS$^{low}$CD122$^+$ population that is T-bet$^+$RORγt$^-$ IFN-γ-producing MAIT1 cells. The relative abundance of MAIT1/17 varies significantly among different organs with MAIT17 cells being predominant in the thymus, LNs, and lung while MAIT1 cells being relatively enriched in the liver and spleen, suggesting that local environment may control MAIT1/17 abundance in different organs. Of note, two most recent reports also used CD122 to identify MAIT1 cells[22]. Our data are not only consistent with these reports but also reveal developmental regulation of this molecules in MAIT cells and provide an approach to clearly define MAIT1 and MAIT17 cells by inclusion of ICOS expression.

ICOS and CD122 not only can serve as surface markers for MAIT1/17 cells but also play differential roles in these effector lineages. Deficiency of ICOS does not obviously affect stages 1 and 2 MAIT cells but leads to a considerable reduction of stage 3 MAIT cells. Within stage 3 MAIT cells, only RORγt$^+$ IL-17-producing MAIT17 cell but not CD122$^+$T-bet$^+$ MAIT1 cell numbers are severely decreased. Although MAIT1 cells express low levels of ICOS, they appear not to rely on ICOS for their differentiation or hemostasis. Thus, ICOS promotes MAIT17 but

not MAIT1 differentiation/maintenance. In cαβT cells and iNKT cells, ICOS enhances the expression of the transcription factor c-MAF and induces mTORC2 activation to promote Th17 and iNKT17 differentiation[47,51,54,69]. Whether ICOS controls cMAF expression to regulate MAIT17 differentiation remains to be investigated. Nevertheless, our data together with these previous observations suggest that ICOS may exert broad roles in IL-17 effector lineage differentiation/homeostasis among different αβT cells. While we are revising the manuscript, another group have also reported that ICOS is highly expressed in MAIT cells and it promotes optimal MAIT cell activation in vivo[26]. Thus, ICOS is important for both MAIT17 differentiation in the thymus and MAIT cell expansion in peripheral organs.

Similar to ICOS deficiency, IL-2/IL-15Rβ deficiency does not cause decreases of stages 1 and 2 MAIT cells, suggesting that IL-2/IL-15R signal does not play an important role for early MAIT cell development, which is consistent with the lack of expression of this receptor in these MAIT cells. However, different from ICOS, both IL-2 and IL-15 signals facilitate MAIT1 differentiation/maintenance but play minimal roles in MAIT17 cell differentiation. Both IL-2 and IL-15 treatments preferentially induce CD122$^+$ MAIT1 cell expansion in vivo and in vitro. Deficiency of IL-2/IL-15Rβ intrinsically impaired T-bet$^+$ MAIT1 cell differentiating/homeostasis without obviously affecting RORγt$^+$ MAIT17 cell numbers. Concordance with IL-2/IL-15Rβ deficiency, T-bet deficient mice lack CD122$^+$ MAIT1 cells but have virtually normal numbers of MAIT17 cells. It has been reported that IL-2/IL-15Rβ and T-bet reciprocally promote each other as IL-15 is able to induce T-bet expression and T-bet, in turn, promotes IL-2/IL-15Rβ expression in iNKT and cαβT cells[71–73]. IL-2 also promotes T-bet expression to enhance Th1 differentiation[74]. The absence of MAIT1 cells when either IL-2/IL-15Rβ or T-bet is deficient suggests that IL-2/IL-15 signal and T-bet may also reciprocally promote each other for MAIT1 differentiation/homeostasis. This notion is further supported by the ability of IL-2 and IL-15 to upregulate T-bet levels in MAIT1 cells in an mTORC1-dependent manner. Because IL-2 and IL-15 are produced by different cells, our data suggest that MAIT1 cells may be subjected to regulations by both conventional T cells and other innate and/or stromal cells in different settings.

TCR-independent cytokine-induced MAIT cell activation has been proposed important for MAIT cell-mediated immune defense against viral and bacterial infection or pathogenesis of diseases. IL-18, IL-12, and type I interferons have been reported to be able to induce MAIT cell activation[15,75–77]. IL-12 and IL-7 also control MAIT cell cytotoxicity[78]. In addition to IL-15-induced MAIT1 expansion, we have revealed that MAIT cells also respond to IL-1β and IL-23. IL-23 selectively acts on MAIT17 cells to promote their survival and, to a less extent, proliferation.

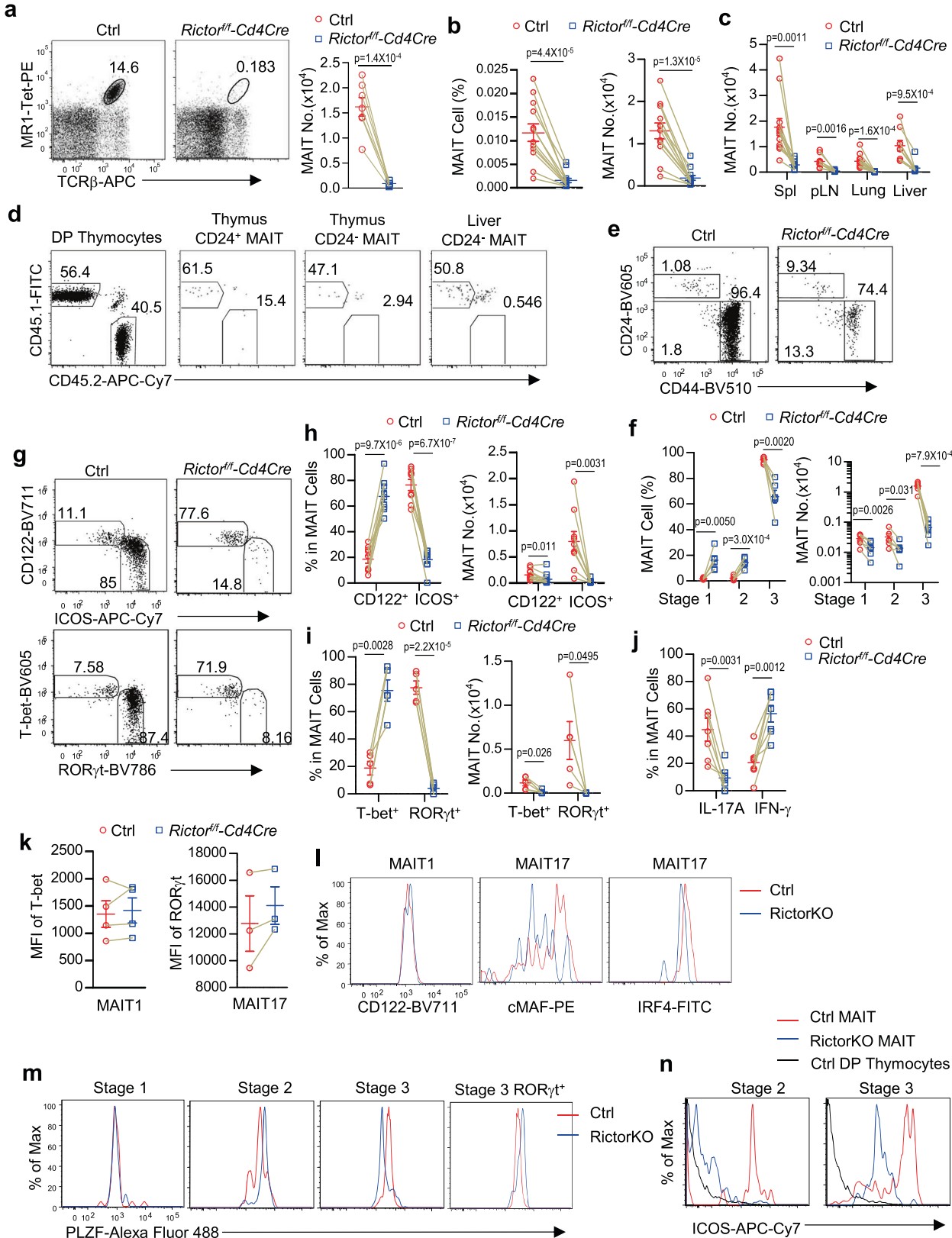

Compared with IL-1β, IL-23 is a weak inducer for MAIT cell expansion, at least in an in-vitro setting. In contrast, IL-1β is less selective than IL-23 and is able to promote both MAIT1 and MAIT17 survival and vigorous proliferation. However, IL-1β does display a stronger influence on MAIT17 cells than MAIT1 cells as it induces more vigorous expansion of MAIT17 cells than

MAIT1 cells. Regardless of these differences, these two cytokines synergistically promote MAIT cell expansion in vitro and, particularly, in vivo. Because these cytokines are induced during infection and inflammatory conditions, signals from their receptors may play important roles in MAIT cell differentiation into different effector lineages and homeostasis of these cells. In

**Fig. 7 Critical role of mTORC2 for MAIT17 effector lineage differentiation.** Eight–ten weeks *Rictor*$^{f/f}$*-Cd4Cre* and *Rictor*$^{f/f}$ *or WT-Cd4Cre* control mice were analyzed for MAIT cells (**a–c**, **e–n**). **a** MR1-Tet and TCRβ staining and MAIT cell numbers after enrichment of thymocytes with 5-OP-RU-loaded MR1-Tet. **b** Thymic MAIT cell percentages and numbers without MR1-Tet enrichment. **c** MAIT cell numbers in peripheral organs. **d** CD45.1$^+$CD45.2$^+$ irradiation recipient mice reconstituted with a mixture of CD45.1$^+$ WT and CD45.2$^+$ *Rictor*$^{f/f}$*-Cd4Cre* BM cells after irradiation were analyzed 8 weeks after reconstitution. Representative dot plots show CD45.1 and CD45.2 staining of CD4$^+$CD8$^+$ DP thymocytes, CD24$^+$ and CD24$^-$ thymic MAIT cells, and CD24$^-$ liver MAIT cells. **e** Representative FACS plots of CD44 and CD24 staining in enriched thymic MAIT cells. **f** Scatter plots show stages 1–3 thymic MAIT cell percentages and numbers after enrichment. **g** CD122 vs ICOS and T-bet vs RORγt staining in live gated Lin$^-$ CD44$^+$ thymic MAIT cells. **h** Scatter plots show CD122$^+$ and ICOS$^+$ MAIT cell percentages and numbers after MR1-Tet enrichment. **i** Scatter plots show T-bet$^+$ and RORγt$^+$ MAIT cell percentages and numbers after enrichment. **j** IFN-γ$^+$ and IL-17A$^+$ MAIT cells after PMA plus ionomycin stimulation. **k** MFI of T-bet and RORγt in MAIT1 and MAIT17 cells. **l** Overlaid histograms show CD122 expression in MAIT1 cells and cMAF and IRF4 expression in MAIT17 cells. **m** PLZF expression in thymic stages 1–3 and RORγt$^+$ stage 3 MAIT cells. **n** ICOS expression in thymic MAIT cells. WT control DP thymocytes were used as control. Each circle and square represents one WT control and mTORC2KO mice, respectively. Connection line represents one pair of WT control and mTORC2KO mice analyzed in one experiment. Data shown are representative of two experiments for (**d**) and are representative of or pooled from three to twelve experiments for all other data. Statistical significance is determined by two-tailed pairwise Student's *t* test. *P* values of less than 0.05 are shown. Source data for all graphs are provided as a Source Data file.

this regard, it has recently been reported that MAIT cells expand rapidly after influenza virus infection to participate in host defense against this pathogen[40]. Most recently, it was also revealed that pulmonary *Legionella* or *Salmonella* infection and administration of multiple TLR ligands together with 5-OP-RU promote MAIT cell expansion in vivo that is dependent on IL-23 and IL-23 is critical for MAIT cell to maintain a MAIT17 profile after bacterial infection[23,26]. Additionally, IL-23 deficient mice have decreased cutaneous MAIT cells and cutaneous IL-1 receptor-deficient MAIT cells manifested decreased IL-17A production upon S. *epidermidis* stimulation[15]. Together, these reports and our data demonstrate that IL-1β and IL-23 are important for MAIT cell, especially MAIT17 cell, homeostasis and expansion. It would be interesting to determine how IL-1β, IL-23, IL-15, and other cytokines are dynamically regulated to control MAIT cell-mediated immune protection, as well as pathogenesis during viral infections and other inflammatory conditions in the future.

T-bet$^+$RORγt$^+$ MAIT cells are virtually undetectable in mice in the steady state. IL-1β and IL-23 treatment induced significant accumulation of T-bet$^+$RORγt$^+$ MAIT cells in vitro and in vivo. Interestingly, pulmonary Legionella and Salmonella infection also induce significant expansion of RORγt$^+$T-bet$^+$ MAIT cells[23,26]. These observations suggest plasticity of MAIT effector lineages under an inflammatory environment. Because these cells are ICOS$^+$CD122$^-$ and resemble MAIT17 cells and could be induced from ICOS$^+$CD122$^-$ MAIT17 cells but not ICOS$^-$CD122$^+$ MAIT1 cells following in vitro IL-1β and IL-23 stimulation, they are most likely derived from T-bet$^-$RORγt$^+$ cells. Future in vivo fate-mapping studies is needed to clearly illustrate the origin(s) and functional features of these cells.

mTOR has the capability to integrate multiple environmental and intracellular cues. In the absence of mTOR, stage 3 MAIT cells were virtually absent but thymic stages 1 and 2 MAIT cell development is not affected or only minimally affected. Thus, mTOR is pivotal for functional maturation of MAIT cells but not for early MAIT cell generation. Interestingly, PLZF or miR-181a/b-1 deficiency also impedes MAIT cell functional maturation to effector cells[29,31], although less severe than mTOR deficiency. Although PLZF protein levels were not obviously decreased in mTOR deficient MAIT cells, it has been reported that mTORC1 promotes PLZF nuclear localization and miR-181-b targets PTEN to promote mTOR signaling[52,79]. The similar impacts of mTOR, PLZF, or miR-181-a/b-1 deficiency on MAIT cell functional maturation and cross-regulation among these molecules suggest a potential miR-181-a/b-1-PTEN-mTOR-PLZF pathway to promote MAIT cell effector lineage differentiation, which warrants further investigation in the future.

We have further demonstrated that deficiency of either mTORC1 or mTORC2 causes severe decreases of MAIT cells in the thymus and peripheral organs. Number-wise, mTORC1 deficiency appears to inflict severer impact on MAIT cells than mTORC2 deficiency. Additionally, mTORC1 and mTORC2 exert differential roles for MAIT effector lineage differentiation. mTORC1 deficiency causes severe decreases of stage 3 MAIT cells, suggesting that mTORC1 is important for stage 3 MAIT cell expansion. Within the remaining thymic MAIT cells in mTORC1 deficient mice, the relative percentages of MAIT1 cells are severely reduced while the relative percentages of MAIT17 cells are not obviously changed, indicating that mTORC1 is particularly essential for MAIT1 differentiation. Interestingly, reduced mTORC1 activity has been observed in MAIT cells from diabetic patients and contributes to their functional impairment of IFN-γ production[61]. Thus, mTORC1 exerts important roles in both human and mouse MAIT cells. It has been reported mTORC1 promotes T-bet phosphorylation and regulates T-cell metabolism to enhance Th1 responses[80,81]. Similar mechanisms may also be involved in MAIT1 differentiation. Given the ability of IL-2 and IL-15 to induced mTORC1 activation and T-bet upregulation in MAIT cells in an mTOR dependent manner, the requirement of mTORC1 for IL-2 and IL-15 induced MAIT1 expansion, the necessity of IL-2/IL-15Rβ, T-bet, and mTORC1 for MAIT1 differentiation, and the importance of mTORC1 for Th1 differentiation and T-bet phosphorylation, we propose an IL-2/IL-15R-mTORC1-T-bet axis in MAIT cells that promote MAIT1 differentiation and/or maintenance.

Different from mTORC1 deficiency, mTORC2 deficiency predominantly inhibits MAIT17 differentiation. MAIT17 cells are virtually absent but the relative percentages of MAIT1 cells are increased in RictorKO mice. Since mTORC2 deficiency also severely impairs iNKT17 cell development[54], MAIT17 and iNKT17 cells share similar requirements of mTORC2 for their development/maintenance. It has been reported that iNKT17 and MAIT17 cells share similar transcriptional programs[82]. mTORC2 phosphorylates and enhances the activities of several molecules, such as SGK1 and Akt. Both SGK1 and Akt can phosphorylate Foxo1 to sequester it from entering the nucleus to function as a suppressor of Th17 responses[83–85]. We have also recently found that Foxo1 suppresses iNKT17 differentiation[86]. mTORC2 may function similarly in MAIT cells to promote MAIT17 differentiation/homeostasis. Because both mTORC2 deficiency and ICOS deficiency preferentially impede MAIT17 differentiation and ICOS promotes mTORC2 activation, we propose that an ICOS–mTORC2 axis may exist in MAIT cells to promote MAIT17 differentiation. Additionally, ICOS levels are decreased

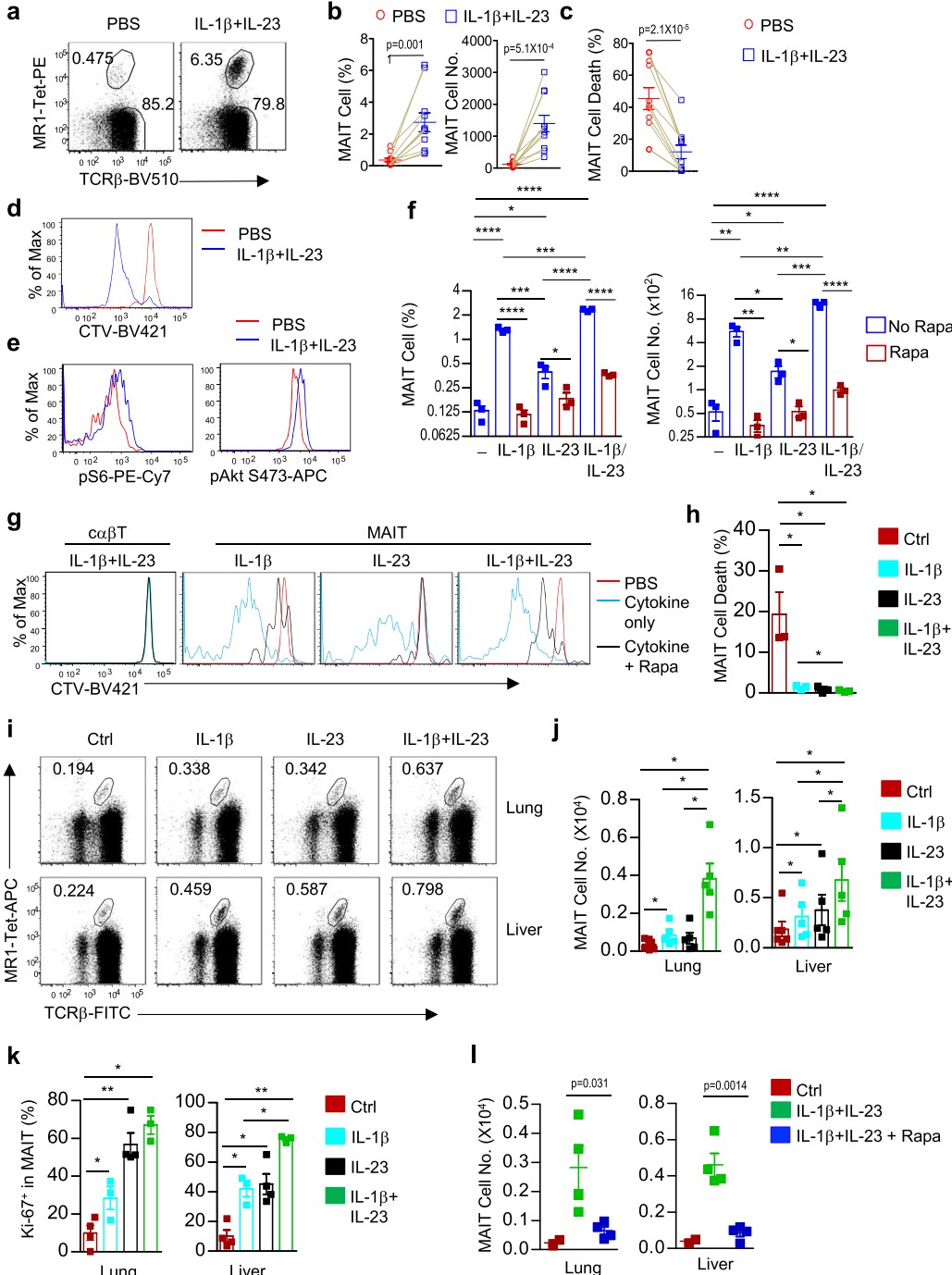

**Fig. 8 IL-1β and IL-23 induced MAIT cell expansion in vitro and in vivo. a–h** Splenocytes from WT mice were treated with IL-1β, IL-23, IL-1β plus IL-23, or PBS directly or after labeled with CTV and examined 72 h after incubation at 37 °C. In some experiments, 20 ng/ml of rapamycin was included. **a** Representative FACS plots of TCRβ and MR1-Tet staining of Lin⁻ cells. **b** MAIT cell percentages and numbers. **c** Death rate of MAIT cells. **d** Overlaid histogram showing MAIT cell proliferation by CTV-dilution. **e** Representative histograms showing S6 and Akt S473 phosphorylation in MAIT cells. **f** MAIT cell percentages and numbers after treatment with IL-1β, IL-23, or IL-1β plus IL-23 in the presence or absence of rapamycin (rapa). **g** MAIT cell proliferation in the presence or absence of rapamycin. **h** Death rate of MAIT cells. **i–k** WT mice were i.p. injected with either IL-1β, IL-23, IL-1β plus IL-23, or PBS on days 1 and 2 and were examined on day 4. **i** Dot plots show TCRβ and MR1-Tet staining in live gated Lin⁻ CD45.2⁺ lung cells or liver mononuclear cells. **j** Bar graphs represent mean ± SEM of liver and lung total MAIT cell numbers. **k** Ki-67⁺ cells in lung and liver MAIT cells. **l** WT mice were i.p. injected with PBS (Ctrl), IL-1β and IL-23, or IL-1β, IL-23, and rapamycin on days 1 and 2 and were examined on day 4. Scatter plots represent mean ± SEM of liver and lung total MAIT cell numbers. **a–e** are representative of or pooled from seven to eleven experiments. Each connection line indicates one experiment. **f–h** are representative of two experiments and bar figures are mean ± SEM calculated from triplicates in one experiment. **i–k** are representative of or pooled from three to five experiments. **l** is representative of two experiments. *$p < 0.05$; **$p < 0.01$; ***$p < 0.001$; ****$p < 0.0001$ or exact $p$ values are determined by two-tailed paired (**b**, **c**, **j**, **k**) and unpaired (**f**, **h**, **l**) Student's $t$ test. Source data for all graphs are provided as a Source Data file.

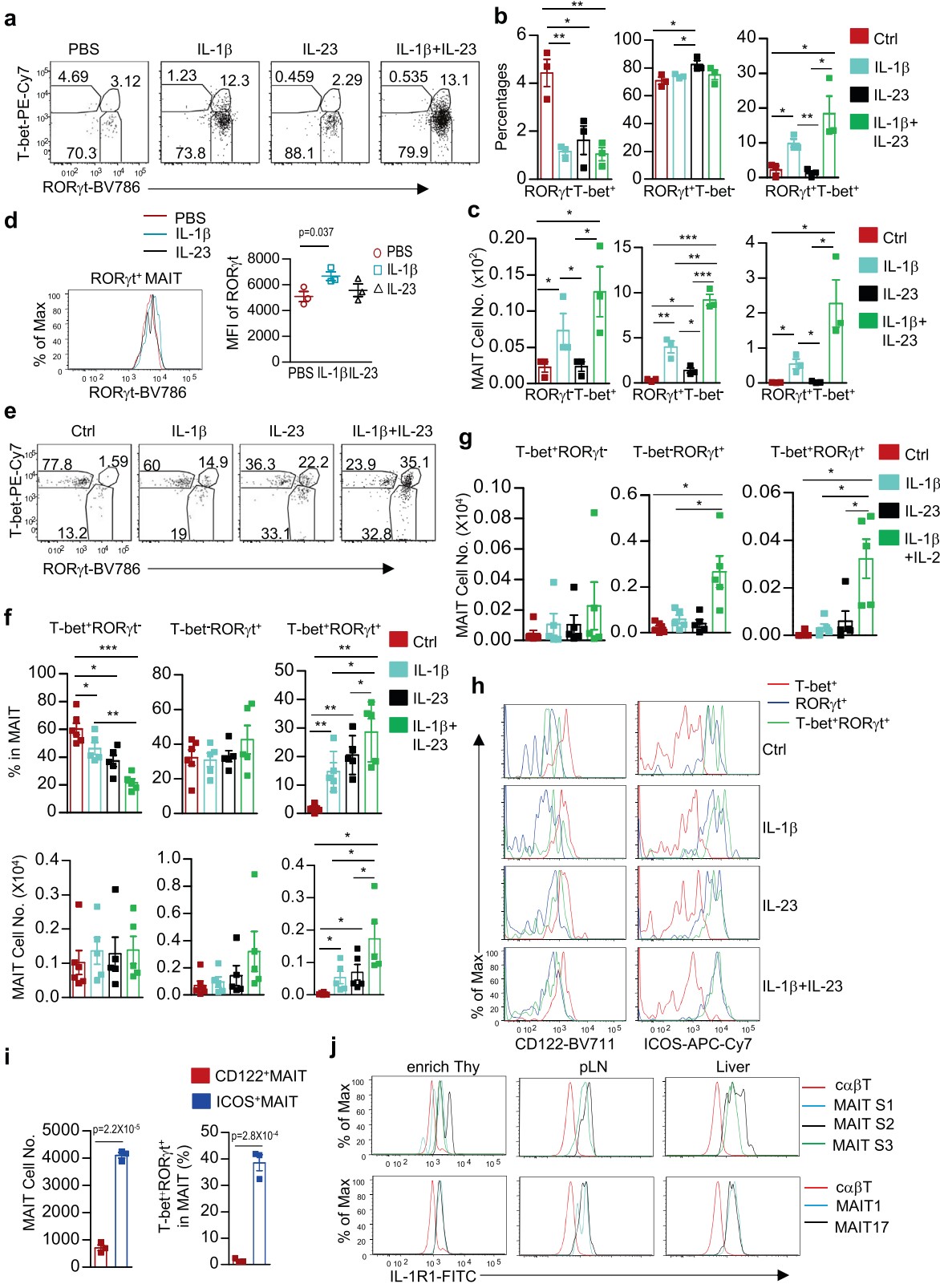

in mTORC2 deficient MAIT cells, suggesting that mTORC2 also promotes MAIT17 differentiation via positive feedback control of ICOS expression. It is interesting to note that cMAF plays critical roles for Th17, γδT17, and iNKT17 lineage differentiation[67–69] and ICOS promotes cMAF expression to control Th17 differentiation[69]. Decreased cMAF but not RORγt expression in

mTORC2 deficient MAIT17 cells suggests possibility that the ICOS-mTORC2 axis might promote cMAF expression to direct MAIT17 differentiation.

Of note, although the mTORC2 and ICOS deficiency lead to similar phenotypes in MAIT cells, decrease of total MAIT cells, particularly MAIT17 cells, in mTORC2 deficient mice is much

**Fig. 9 IL-1β and IL-23 influenced MAIT cell effector lineages in vitro and in vivo. a–d** Splenocytes from WT mice were treated with IL-1β, IL-23, IL-1β plus IL-23, or PBS and examined 72 h after incubation at 37 °C. **a** T-bet and RORγt staining in MAIT cells. **b, c** Scatter plots showing RORγt⁻T-bet⁺, RORγt⁺T-bet⁻, and RORγt⁺T-bet⁺ MAIT cell percentages (**b**) and numbers (**c**). **d** Overlaid histogram showing RORγt levels in RORγt⁺ MAIT17 cells. Scatter plot shows MFI of RORγt. **e–h** WT mice were i.p. injected with either IL-1β, IL-23, IL-1β plus IL-23, or PBS on days 1 and 2 and were examined on day 4. **e** T-bet and RORγt staining in liver MAIT cells. **f** T-bet⁺RORγt⁻ MAIT1, T-bet⁻RORγt⁺ MAIT17, and T-bet⁺RORγt⁺ MAIT cell percentages and numbers in the liver. **g** T-bet⁺RORγt⁻ MAIT1, T-bet⁻RORγt⁺ MAIT17, and T-bet⁺RORγt⁺ MAIT cell numbers in the lung. **h** CD122 and ICOS levels in the indicated liver MAIT cell populations. **i** MAIT cell numbers and percentages of T-bet⁺RORγt⁺ cells after stimulation of sorted CD24⁻CD44⁺ICOS⁻CD122⁺ MAIT1 and CD24⁻CD44⁺ICOS⁺CD122⁻ MAIT17 cells with IL-1β and IL-23 for 3 days. **j** Overlaid histograms showing ex vivo assessment of IL-1R1 expression in cαβT cells, stages 1, 2, and 3 MAIT cells (top panels), as well as in T-bet⁺ MAIT1 and RORγt⁺ MAIT17 cells (bottom panels) in the indicated organs from WT mice. **a–d** are representative of two experiments and bar figures are mean ± SEM calculated from triplicates in one experiment. **e–h** are representative of or pooled from four to five experiments. **i** is mean ± SEM of triplicates of one experiment and represents two experiments. **j** is representative of three experiments. *, $p < 0.05$; **, $p < 0.01$; ***, $p < 0.001$ or exact p values are determined by two-tailed unpaired (**b, c, d, i**) and pairwise (**f, g**) Student's t test. Source data for all graphs are provided as a Source Data file.

severe than in $Icos^{-/-}$ mice. Given the ability of IL-1β and IL-23 to induce mTOR activation in MAIT cells, the strong effector of these cytokines on MAIT17 cells, and the requirement of DAG mediated signaling for efficient MAIT17 differentiation, we propose that mTORC2 may also integrate IL-1β/IL-23 signal and possibility other signals, such as TCR signal in addition to ICOS to exert its crucial role for MAIT17 differentiation.

## Methods

**Mice.** $Icos^{-/-}$ (Stock No: 004859), $Raptor^{f/f}$ (Stock No: 013188), $mTOR^{f/f}$ (Stock No: 011009), $Rictor^{f/f}$ (Stock No: 020649), $Il2/15rb^{f/f}$ (Stock No: 029657), $Tbx21^{f/f}$ (Stock No: 022741), and $Cd2iCre$ (Stock No: 008520) mice were purchased from the Jackson laboratory and backcrossed to C57BL/6J background. $Cd4Cre$ mice (Model No: 4196) were originally purchased from Taconic Inc. $TCRJa18^{-/-}$ mice[87] were kindly provided by Drs. Masaru Taniguchi, Kim Nichols, and Luc Van Kaer. $Dgkz^{WT}$-$Cd4Cre$ and $Dgkz^{ΔNLS}$-$Cd4Cre$ mice with T cell-specific expression of WT-DGKζ or a gain-of-function DGKζ$^{ΔNLS}$ mutant were recently reported[27,62]. Primers for mouse genotyping are listed in supplementary Table 1. All animals were housed in specific-pathogen-free conditions. Experimental and control animals were co-housed together and were euthanized with carbon dioxide followed by organ removal. Both male and female mice were studied at 2–4 months of age. Experiments described were approved by the Institutional Animal Care and Use Committee of Duke University.

**Preparation of single-cell suspension and enrichment of MAIT cells.** Single-cell suspensions of the thymus, spleen, LNs, lung, and liver mononuclear cells (MNCs) were prepared as previously described[88,89]. MAIT cell enrichment with MR1-Tet and MACS beads from total thymocytes was described in detail in a published report[66]. For MAIT cell analysis, single-cell suspensions with or without MR1-Tet enrichment were stained with anti-TCRβ, CD24, CD44, and other antibodies and, for unenriched cells, PE or APC-conjugated MR1-Tet. Lineage markers, CD19, PBS-57-loaded CD1d-Tet, TCRγδ, CD11b, CD11c, F4/80, B220, Gr1, and Ter119 were costained for the exclusion of other cell lineages. LIVE/DEAD Fixable Dead Cell Stain was used to exclude dead cells. MAIT cells were gated on singlet live Lin⁻ TCRβ⁺MR1-Tet⁺ cells in the thymus and on singlet live Lin⁻ CD24⁻ TCRβ⁺MR1-Tet⁺ cells in other organs (Supplementary Fig. 10).

**Antibodies and flow cytometry.** Fluorochrome-conjugated anti-CD45.2 (clone 104), CD45.1 (clone A20), ICOS (clone C398.4A), CD4 (clone GK1.5), CD8 (clone 53-6.7), TCRβ (clone H57-597), TCRγδ (clone GL3), CD44 (clone IM7), CD24 (clone M1/69), Gr1 (clone RB6-8C5), CD11b (clone M1/70), CD11c (clone N418), F4/80 (clone BM8), CD19 (clone 6D5), MR1 Tetramer (NIH tetramer facility), CD1d Tetramer (NIH tetramer facility), B220 (clone RA3-6B2), Ter119/Erythroid Cells (clone TER-119), Streptavidin, T-bet (clone 4B10), IL-1R1 (clone JAMA-147), IFN-γ (clone XMG1.2), and IL-17A (clone TC11-18H10.1) were purchased from Biolegend unless indicated otherwise. CD122 (clone TM-Beta1), RORγt (clone Q31-378), Ki-67 (catalog number: 550609), CD45 (clone 30-F11), and PE Isotype Control (clone MOPC-21) were purchased from BD Biosciences. Phospho-S6 (Ser235/236, cupk43k) and phosphor-Akt Ser473 (SDRNR) were purchased from eBioscience. Phosphor-Akt T308 (244F9), Raptor (24C12), and Rictor (53A2) antibodies were from Cell Signal. Alexa Fluor 568 goat anti-mouse IgG (ThermoFisher, catalog number: A11031). Texas Red goat anti-rabbit IgG (Molecular Probes, catalog number: T-2767). PLZF (clone Mags.21F7), IRF4 (clone 3E4), BATF (clone MBM7C7), and cMAF (clone symOF1) were purchased from Thermofisher. Cells were stained for cell surface molecules using 2% FBS-PBS. Cell death was identified by using the Violet Live/Dead cell kit (Thermofisher), Fixable Viability Dye eFluor 780, or 7-AAD. Intracellular staining for RORγt, T-bet,

cMAF, IRF4, BATF, and PLZF was performed using the eBioscience Foxp3 Staining Buffer Set. Intracellular staining of phosphor-S6, phosphor-Akt, phosphor-Akt T308, Raptor, and Rictor was performed by using the BD Biosciences Cytofix/Cytoperm and Perm/Wash solutions. Stained samples were acquired on a FACS Canto-II or a BD LSRFortessa™ (BD Biosciences) flow cytometer. Data were analyzed with FlowJo software (Tree Star). Detailed antibody information is summarized in Supplementary Table 2.

**Stimulation and intracellular cytokine detection.** MR1-Tet enriched thymocytes or liver MNCs were stimulated with phorbol 12-myristate 13-acetate (PMA, 50 ng/ml) plus ionomycin (500 ng/ml) in the presence of GolgiPlug (1 ng/ml) for 4–5 h. After cell surface staining, intracellular staining for IL-17A and IFN-γ was performed by using the BD Biosciences Cytofix/Cytoperm and Perm/Wash solutions. To stimulate ICOS, liver mononuclear cells in 10%-FBS-IMDM were incubated with an anti-ICOS antibody (5 μg/ml; clone C398.4A; Invitrogen) at 37 °C for 30 min, followed cell surface and intracellular staining of MAIT cells.

**IL-15/IL-15RαFc, IL-2, IL-23, and IL-1β treatment.** IL-15/IL-15RαFc complex was prepared by mixing an equal volume of 10 μg/ml IL-15 (PEPROTECH, catalog number: 210-15) and 50 μg/ml IL-15Rα/Fc (R&D Systems, catalog number: 551-MR), followed by incubation at 37 °C for 30 min. IL-2/anti-IL-2 antibody complex was similarly prepared by mixing 1 μg mouse IL-2 (PEPROTECH, catalog number: 212-12) and 5 μg anti-IL-2 antibody (clone S4B6-1, BioXCell, catalog number: BE0043-1) in 200 μl PBS[64] followed by incubation at 37 °C for 30 min. For in vivo treatment, WT mice were i.p injected with 200 μl of either the IL-2 /anti-IL-2 complex (6 μg/day) on days 1, 2, and 3, the IL-15/IL-15RαFc complex (12 μg/mouse) on day 2, or both and euthanized for experiment on day 5. For IL-1β and IL-23 treatment, WT mice were i.p. injected with either IL-1β (1 μg/day, PEPROTECH, catalog number: 211-1B), IL-23 (1 μg/day, R&D, catalog number: 1887-ML-010), or both with or without coinjection of 75 μg/kg body weight rapamycin on days 1 and 2 and euthanized for experiment on day 4. For in vitro treatment, unlabeled or CellTrace™ Violet (CTV, ThermoFisher Scientific) labeled splenocytes from WT mice were resuspended in 10% FBS containing IMDM supplemented with 55 uM 2-ME, 1 mM L-glutamine, 100 U/ml penicillin, and 100 mg/ml streptomycin (IMDM-10), seeded in 96-well U-bottom plates ($3 \times 10^6$ cells per well in 200 μl IMDM-10), and cultured in the absence or presence of IL-15/IL-15RαFc complexes (0.05 ug/ml IL-15/0.25 ug/ml IL-15RαFc final concentration), IL-2 (1000 ng/ml), IL-1β (10 ng/ml), IL-23 (10 ng/ml), or 10 ng/ml IL-1β plus 10 ng/ml IL-23 at 37 °C, 5% $CO_2$ for 3 days before harvest for analysis. In some experiments, 20 ng/ml rapamycin was also added to the culture.

**In vitro stimulation of and detection of cytokines from sorted MAIT cells.** CD24⁻CD44⁺ CD122⁺ICOS^low and CD122⁻ICOS⁺ MAIT cells were sorted from liver MNCs from WT mice. Three thousand sorted MAIT cells were stimulated with PMA (50 ng/ml) and ionomycin (500 ng/ml) in 200 μl 10% FBS-IMDM in a 96-well plate at 37 °C for 4 h. Cytokines in culture supernatants were measured using the LEGENDplex™ MU Th Cytokine Panel with VbP V03 kit (Biolegend, Cat: 741044) with a slight modification of shaking the plate at 150 rpm. Standards and samples were acquired on a BD LSRFortessa™ flow cytometer. Data were analyzed with the LEGENDplex™ Data Analysis Software. Additionally, 1500 sorted MAIT1 and MAIT17 cells were stimulated with 10 ng/ml IL-1β plus 10 ng/ml IL-23 at 37 ºC, 5% $CO_2$ for 3 days, followed by cell surface TCRβ and MR1-Tet staining and intracellular anti-T-bet and anti-RORγt staining and FACS analysis.

**Assessment of TCRVα-Jα recombination.** TCRVα-Jα recombination was performed according to a published protocol with modifications[89]. Briefly, $1 \times 10^7$ viable CD4⁺CD8⁺ thymocytes from age- and sex-matched WT and $mTOR^{f/f}$-$Cd4Cre$ mice were sorted on MoFlo Cell Sorter (Beckman Coulter) with post sort purity of 98%. Genomic DNAs were extracted with phenol/chloroform,

precipitated with 70% ethanol, and dissolved in TE buffer (10 mM Tris-0.5 mM EDTA [pH 8]). For semi-quantitative PCR, decreasing amounts of DNA template (100, 33, and 11 ng) from each sample were used. The forward primer for TCRVα19 segment was 5′-GTACAGTTACCTGCTTCTGAC-3′. The reverse primers for different TCRJα segments were as follows: TCRJα18, 5′-GTAGAAAGAA ACCTACTCACCA-3′, and TCRJα33, 5′-TATCGAAGACTTACCTGGCTT-3′. Primers for Cd14 PCR (loading control) were 5′-GCTCAAACTTTCAGAA TCTACCGAC-3′ (forward) and 5′-AGTCAGTTCGTGGAGGCCGGAAATC-3′ (reverse).

**Analyses of scRNAseq data**. Raw counts of scRNAseq data reported by Legoux et al[30] were downloaded from European Bioinformatics Institute (EMBL-EBI) under the accession number E-MTAB-7704 (single-cell RNA-seq). scRNAseq data were pre-processed using the Seurat package (version 3.1.1) in R (version 3.5.3)[90]. Genes expressed in less than three cells and cells with no more than 50 detected genes were filtered out. Filtered datasets were normalized the gene expression measurements for each cell by the total expression multiplied with a scale factor of 10,000 by default, followed by log-transformation of the results using the global-scaling normalization method, LogNormalize. The technical noise and (or) biological sources of variation were mitigated via ScaleData function to improve downstream dimensionality reduction and clustering. Highly variable genes were screened with FindVariableFeatures function for downstream analysis. Principle component analysis (PCA) was performed on the scaled data using the RunPCA function. Significant PCs were identified as those with a strong enrichment of low $p$ value genes based on the Jackstraw algorithm. For cell clustering, k-nearest neighbors were calculated and the SNN graphs were constructed using Find-Neighbors. Top 20 PCs were selected for analysis using FindClusters. Cells within the graph-based clusters determined above were co-localized for visualization on the tSNE plot via RunTSNE and TSNEPlot. FindAllMarkers were applied to find markers that define clusters via differential expression. FeaturePlot was applied to visualize individual gene expression on a tSNE plot. VlnPlot was applied to show expression probability distributions across clusters.

**Statistical analysis**. The paucity of MAIT cells in mice introduces variations between experiments. To overcome this issue, we performed individual experiments examining a pair of age- and sex-matched test and control mice with most pairs being littermates and housed in the same cage. Each pair of mice in the individual experiment was marked by a connecting line between test and control mice. Scatter plots in each figure were pooled from multiple experiments and the numbers of pairs showed in the plots reflect the numbers of experiments performed. Comparisons were made using two-tailed pairwise Student's $t$ test using the Prism 5/GraphPad software. For experiments that did not fall into the pairwise category, comparisons were made using two-tailed unpaired Student's $t$ test. Data shown are presented by mean ± SEM. Only $p$ values less than 0.05 are considered significant and are shown in figures.

**Reporting summary**. Further information on research design is available in the Nature Research Reporting Summary linked to this article.

## Data availability

All relevant data are available from the corresponding author. Source data are provided with this paper.

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

## Acknowledgements

We thank the NIH Tetramer Facility for providing MR1- and CD1d-Tetramers, the Flow Cytometry Facility in Duke Cancer Institute for cell sorting, and Jeffrey Zhong for editing the manuscript. Research in this manuscript has been supported by NIH (R01AI079088, R01AI101206, R56AG060984, and R56AI079088).

## Author contributions

H.T., Y.P., C.S., L.L., J.X., P.W., S.Z., and S.R. designed and performed experiments, analyzed data. H.T., Y.P., and S.Z. prepared the figures. H.T. and L.L. contributed to manuscript preparation. J.W.S. participated in data interpretation and manuscript preparation. X.-P.Z. conceived the project, designed experiments, participated in data analysis, and wrote the paper.

## Competing interests

The authors declare no competing interests.
