## [Peer Review File · Nature Communications]

Reviewers' comments:

Reviewer #1 (Th polarization, cytokine signaling, immune cell transcription regulation)(Remarks to the Author):

In the current paper, the authors have described the mechanisms underlying MAIT (MAIT1 and MAIT17) cells development and effector lineage differentiation. The paper covers functional markers/factors that are critical for MAIT1 (IL-15R, mTORC1, and Tbet) and MAIT17 (mTORC2, ICOS, IL-1bR/IL-23R, and ROR γ t) cell differentiation. While the biology of MAIT cells has been described and some of the key points presented in this paper have already been shown in several publications, for instance, the role of mTORC pathway and IL-15 signaling has been shown to participate in regulation of MAIT cells (O'Brien A1, Loftus RM2, Pisarska MM, et al. J Immunol. 2019 Jun 15; 202 (12):3404-3411; Sattler A1, Dang-Heine C1, Reinke P, et al. Eur J Immunol. 2015 Aug; 45 (8):2286-98; Legoux F1, Gilet J2, Procopio E, et al. Nat Immunol. 2019 Sep; 20 (9):1244-1255); further understanding of MAIT cell development and biology is important since MAIT cells play a significant role in host defense and are involved in inflammatory disorders. Though the paper is well structured and the main hypothesis is clearly stated some major and minor concerns must be counseled prior to the publication. In particular, if the authors describe more comprehensively and deeper the proposed signaling pathways essential for MAIT cell development it will contribute significantly to the understanding of MAIT cells biology.

Major concerns:

1. For Figure 1, we suggest more extensive characterization of MAIT1 and MAIT17 cell phenotype. The characterization of these cells is only limited to cytokine (IL-17 and IFN γ) and transcriptional factor (Tbet and ROR γ t) expression. We strongly recommend to examine the additional cytokines (MAIT1: TNF α and Granzyme B; MAIT17: IL-17F, IL-22, IL-21, IL-10 and GM-CSF) and factors (MAIT1: STAT1 and STAT4; MAIT17: STAT3, Batf, IRF4 and c-Maf) that could be essential for MAIT1 and MAIT17 lineage commitment.
2. Figure 2 describes the role of IL-15 signaling in MAIT1 cell development. Since the beta subunit of IL15R is common for both IL-2 and IL-15 receptor signaling, it will be important to determine the role of IL-2 signaling in MAIT1 cell development as well.
3. Figure 3 is dedicated to address the role of ICOS in MAIT17 cell development. However, it is unclear if ICOS signaling is critical for MAIT17 lineage commitment because authors did not use ICOSL or ICOS agonistic antibodies for MAIT cell differentiation. In addition, the source of ICOSL for MAIT lineage commitment (B, DCs, M ϕ s or..) is unclear. These points need to be addressed.
4. In Figure 4, authors show the contribution of mTORc signaling pathway in stage 3 MAIT cells development. We encourage the authors to include analysis for AKT Thr 308, as well as Rictor and Raptor proteins upon IL-15 pathway stimulation. Moreover, the authors should show whether ICOS signaling in MAIT cells contributes to mTORc pathway activation by using ICOS ligand or ICOS agonistic antibodies.
5. For Figure 5, the authors went further to describe the role of mTORc pathway in MAIT cell development and proposed mTORC1 pathway as a key regulator of MAIT1 cell lineage commitment. We recommend to analyze cytokine expression by Raptor deficient MAIT cells (IFN γ , TNF α , Granzyme B, IL-17A, IL-17F, IL-22, IL-21, IL-10 and GM-CSF). Moreover, to prove the role of IL-15R-mTORC1-Tbet axis in MAIT1 cell differentiation, Raptor deficient MAIT cells have to be analyzed after IL-15 pathway activation.
6. For Figure 6, we recommend to design experiment which will further show crosstalk between mTORc2 and ICOS: either ICOS induce mTORc2 or mTORc2 signaling triggers ICOS expression or autocrine regulation of these pathways. In addition, it will be interesting to determine the role of TCR signaling in ICOS expression in MAIT17 cells. Moreover, we recommend to analyze cytokine expression by Rictor deficient MAIT cells (IFN γ , TNF α , Granzyme B, IL-17A, IL-17F, IL-22, IL-21, IL-10

and GM-CSF).

7. In Figure 7, authors tried to address the role of IL1 β and IL-23 in MAIT cell biology. Based on results in Figure 7I, IL1 β signaling but not IL-23 can lead to development of double-positive T-bet+ROR γ t+ MAIT cell population, suggesting the role of IL-1 β in MAIT17 and MAIT1 cell plasticity. We suggest to answer this question by activating sorted MAIT1 or MAIT17 cells with IL-1 β . In addition it will be interesting to determine the role of IL-12 for MAIT1 development and TGF β , IL-6, and IL-21 for MAIT17 cell differentiation as well as to check the expression level of receptors for IL-21, IL-6, and IL-23 cytokines.

Minor concerns:

1. In Figure 6G please include the percentage for the ICOS+ population.
2. In Introduction, we recommend to extend the Introduction by including the following key aspects of MAIT cell biology: the limited array of TCR β -chain, TCR-dependent MAIT cell activation through the TLR pathway, and the cytokine profile of MAIT cells.

Reviewer #2 (MAIT, MR1, T cell biology)(Remarks to the Author):

General comments:

Using a number of genetically modified mice and an in vitro MAIT cell activation assay, the authors firstly identified CD122 and ICOS as potential surrogate cell surface markers to define MAIT cell phenotypes: MAIT1 and MAIT17 respectively. A functional role for CD122 and ICOS on MAIT cell differentiation was also observed. Secondly, signalling via IL15R, IL1betaR and IL23R were found important for MAIT cell proliferation and phenotypic differentiation (MAIT1 and MAIT17). The authors then examined the downstream signalling components: Raptor and Rictor in the mTOR complex that shape the MAIT1 and MAIT17 polarisation. The authors concluded that mTORC2 integrates signals from ICOS and IL1 β R/IL23R to exert a crucial role for MAIT17 differentiation and an IL15R-mTORC1-T-bet axis ensures MAIT1 differentiation.

The design of the experiments is appropriate to the aims but in some instances amplifies the reported findings of others, rather than representing breakthroughs. For example, recent reports by the Lantz group described CD122 as a phenotypic marker (Legoux et al., Nature immunology 2019 20(9):1244-1255) and Legoux et al., Science 10.1126/science.aaw2719 (2019). The transcriptional upregulation of IL15 is flagged in the papers in Cell Reports (2019) from the Klenerman and McCluskey groups where Leng et al also describe the importance of IL-15 in expanding MAIT1 cells in vitro. A role for IL15 in MAIT1 regulation has also reported by Sattler et al, Eur J Immunol. 2015 Aug;45(8):2286-98.

Several groups at the recent Oxford conference on CD1-MR1 reported a key role for ICOS-ICOSL interactions in MAIT cell development and activation though these are not yet published.

As acknowledged by the authors, the function of the mTOR complex on differentiation towards MAIT1 and MAIT17 has been well described in NKT1 and NKT17 differentiation (Wei, Yang, & Chi, 2014) despite the finding is novel in the field (MAIT biology).

The data set from mTOR conditional KO mice is novel in the field (MAIT biology) and will fill a knowledge gap.

1. Some findings may be nonspecific (and not physiological), therefore should be validated in vivo.

The observations on the role of IL1beta and IL23 from in vitro experiments (Figure 7) may not be physiological and it is very important that this is verified by in vivo experiments. For instance, a critical activation role of IL18 has been reported in in vitro studies but could not be reproduced by in vivo experiments (Jesteadt et al., 2018).

2. For Figure 4L: A similar study on iNKT cells, published in PNAS (Shin et al 2014) from Zhong lab, showed that although mTOR has no effect on total expression of PLZF, it controls the nuclear localization of PLZF, which is essential for iNKT-cell lineage development and effector function. The authors should ascertain whether this also occurs in MAIT cells.

Minor issues:

1. It is important to show the total MAIT cell numbers in combination with MAIT cell frequency (%), as there seemed an unstated assumption of a binary composition of MAIT cells into (MAIT1 and MAIT 17). In reality, there is a significant proportion of MAIT cells with mixed transcriptional and functional differentiation. Notably too, the frequency and/or absolute number of MAIT1 versus MAIT17 cells may be not suggest identical shifts in some situations where other cell types are shifting as well.

- Figure 2J showed that total MAIT cell number decreased in several organs in IL2/15Rb KO mice and Figure 2L shows that the MAIT17 frequency was not altered in these organs except for spleen. Did these observations indicate that absolute MAIT17 number decreased as well in this KO mice? Also, Figure 2B showed a significant increase of MAIT17(CD122-) in culture in the presence of IL15. Therefore, IL15 signals are predicted to affect MAIT17 cells, in addition to MAIT1 cells (the main point)?

- Figure 5G and 5H have the same issue as Figure 2L. Absolute number of MAIT17 cells should be shown as well.

2. Figure 4I: this lacks a negative control from MR1-/- mice. Also, in the section of materials and methods the authors did not mention which anti-MR1 antibody was used.

3. Some citations are not accurate. For instance, Line 64 "...display cytotoxic activity^{10, 23, 24}". Reference 10 did not mention granzyme B or cytotoxic activity.

4. The referencing is missing some key papers, e.g. Chen et al Mucosal Immunology 10(1):58-68 describes the MAIT1/MAIT17 polarisation in vivo during infectious challenge.

Point-by-point responses to reviewers' comments

Reviewer #1 (Th polarization, cytokine signaling, immune cell transcription regulation)

In the current paper, the authors have described the mechanisms underlying MAIT (MAIT1 and MAIT17) cells development and effector lineage differentiation. The paper covers functional markers/factors that are critical for MAIT1 (IL-15R, mTORC1, and Tbet) and MAIT17 (mTORC2, ICOS, IL-1bR/IL-23R, and ROR γ t) cell differentiation. While the biology of MAIT cells has been described and some of the key points presented in this paper have already been shown in several publications, for instance, the role of mTORC pathway and IL-15 signaling has been shown to participate in regulation of MAIT cells (*O'Brien A1, Loftus RM2, Pisarska MM, et al. J Immunol. 2019 Jun 15; 202 (12):3404-3411; Sattler A1, Dang-Heine C1, Reinke P, et al. Eur J Immunol. 2015 Aug; 45 (8):2286-98; Legoux F1, Gilet J2, Procopio E, et al. Nat Immunol. 2019 Sep; 20 (9):1244-1255*); further understanding of MAIT cell development and biology is important since MAIT cells play a significant role in host defense and are involved in inflammatory disorders. Though the paper is well structured and the main hypothesis is clearly stated some major and minor concerns must be counseled prior to the publication. In particular, if the authors describe more comprehensively and deeper the proposed signaling pathways essential for MAIT cell development it will contribute significantly to the understanding of MAIT cells biology.

Response: We appreciate the reviewer for providing the insights and references. We have referenced these important studies in our manuscript. In our manuscript, we examined eight genetically manipulated mouse models (including two additions for the revision) and extensively examined each mouse model. While we have tried our best to address all the recommended experiments for many of the mouse models, we were not able to obtain all the data as MAIT cells are very rare in mice and are extremely rare in some of our models. We hope that the reviewer would agree with us our revised manuscript is much improved and some of the recommended experiments can be explored in the future as our data contribute significantly to MAIT cell biology. We also believe that our data of mTOR signaling are novel as there is no paper published experimentally examined the role of mTOR in MAIT cell development and effector lineages differentiation.

Major concerns:

1. For Figure 1, we suggest more extensive characterization of MAIT1 and MAIT17 cell phenotype. The characterization of these cells is only limited to cytokine (IL-17 and IFN γ) and transcriptional factor (Tbet and ROR γ t) expression. We strongly recommend to examine the additional cytokines (MAIT1: TNF α and Granzyme B; MAIT17: IL-17F, IL-22, IL-21, IL-10 and GM-CSF) and factors (MAIT1: STAT1 and STAT4; MAIT17: STAT3, Batf, IRF4 and c-Maf) that could be essential for MAIT1 and MAIT17 lineage commitment.

Response: We have analyzed scRNAseq data of thymic MAIT cells recently published by Dr. Lantz's group because they sequenced large numbers of MAIT cells. These data provide further validation for the utility of ICOS and CD122 as markers to define MAIT1 and MAIT17 lineages and reveal the patterns of additional cytokines in these MAIT subsets. scRNAseq data also reveal that MAIT17 cells also express

transcription factor cMaf, several AP-1 family transcription factors, and IRF4. We have also detected cMaf, IRF4, and Batf at the protein level and found that these transcription factors are expressed at higher levels in MAIT17 cells than in MAIT1 cells. These data are shown in Figure 1G and 1H and supplemental Figure S2. We have tried to assess expression of additional cytokines in these MAIT cells by intracellular staining. However, we have not been able to obtain conclusive data. We have also assessed phosphor-Stat1, -Stat3, and -Stat4 and did not see obvious differences between MAIT1 and MAIT17. However, we are not confident enough to make a conclusion about these data. We believe that it will require more intensive and in-depth studies about this issue. For this reason, we do not include the p-Stat1/3/4 data in the revised manuscript.

2. Figure 2 describes the role of IL-15 signaling in MAIT1 cell development. Since the beta subunit of IL15R is common for both IL-2 and IL-15 receptor signaling, it will be important to determine the role of **IL-2 signaling** in MAIT1 cell development as well.

Response: We have performed the experiments in vitro and in vivo. We have found that IL2 induced MAIT1 cell expansion in vitro and in vivo and upregulated T-bet levels in MAIT1 cells. These new data are shown in Figure 2I – 2O. We have also found that IL2 induced mTOR activation and its effects on MAIT cell expansion was inhibited by rapamycin. These new data are shown in Figure 5A and 5B. Because the inclusion of new data, we separated data about IL2/15rb deficient mice from previous figure 2 into new figure 3.

3. Figure 3 (new figure 4) is dedicated to address the role of ICOS in MAIT17 cell development. However, it is unclear if ICOS signaling is critical for MAIT17 lineage commitment because authors did not use ICOSL or ICOS agonistic antibodies for MAIT cell differentiation. In addition, the source of ICOSL for MAIT lineage commitment (B, DCs, Mφs or..) is unclear. These points need to be addressed.

Response: We have stimulated MAIT cells with an agonist anti-ICOS antibody. Such stimulation was able to induce mTOR activation in MAIT cells. We have included the data in new figure 5C. However, we do not see obvious alteration of MAIT cell numbers after anti-ICOS stimulation in vitro, suggesting ICOS signal by itself is not sufficient to induce MAIT cell activation. We believe that this is an interesting issue that requires further in depth study in the future. More recently, Drs. Chen, Corbett, and Strugnell's groups have also reported that bacterial induced MAIT cell expansion in the lung from ICOS deficient mice are impaired. With regard to the source of ICOS-L for engaging with MAIT cells, we agree that it is an important question. However, it will take tremendous efforts to pinpoint its source and beyond the scope of the current report. I hope that the reviewer would agree with us. Additionally, as pointed out by reviewer 2, other groups are working on this issue and have presented in a scientific meeting.

4. In Figure 4 (new figure 5), authors show the contribution of mTORc signaling pathway in stage 3 MAIT cells development. We encourage the authors to include analysis for **AKT Thr 308, as well as Rictor and Raptor proteins upon IL-15 pathway stimulation.** Moreover, the authors should show whether ICOS signaling in MAIT cells contributes to mTORc pathway activation by using ICOS ligand or ICOS agonistic antibodies.

Response: We have performed the experiments as advised. We have found that IL15 stimulation induced Akt T308 phosphorylation in MAIT cells, that IL2 induced mTORC1, mTORC2 and PI3K activation in MAIT cells, that both IL2 and IL15 stimulation weakly upregulated Raptor and Rictor expression in MAIT cells, and that IL2 also induced MAIT cell proliferation in an mTOR dependent manner. The data are shown in Figure 5A – 5B. Additionally, we have found that stimulation of ICOS with an agonistic antibody can induce mTORC1, mTORC2, and weak PI3K activation in MAIT cells. The data are shown in Figure 5C.

5. For Figure 5 (new Figure 6), the authors went further to describe the role of mTORc pathway in MAIT cell development and proposed mTORC1 pathway as a key regulator of MAIT1 cell lineage commitment. We recommend to analyze cytokine expression by Raptor deficient MAIT cells (*IFN γ* , *TNF α* , *Granzyme B*, *IL-17A*, *IL-17F*, *IL-22*, *IL-21*, *IL-10* and *GM-CSF*). Moreover, to prove the role of IL-15R-mTORC1-Tbet axis in MAIT1 cell differentiation, Raptor deficient MAIT cells have to be analyzed after IL-15 pathway activation.

Response: There are very few MAIT cells in RaptorKO mice. We regret that we are not able to generate clear data to confidently conclude how expression of these cytokines and granzyme B is affected. However, we have showed that rapamycin treatment inhibited IL2 and IL15 induced MAIT1 cell expansion and T-bet upregulation in MAIT1 cells, supporting an IL2/15R-mTORC1-Tbet axis in MAIT1 cell differentiation. These new data are shown in Figure 6J.

6. For Figure 6 (new Figure 7), we recommend to design experiment which will further show crosstalk between mTORc2 and ICOS: either ICOS induce mTORc2 or mTORc2 signaling triggers ICOS expression or autocrine regulation of these pathways. In addition, it will be interesting to determine the role of TCR signaling in ICOS expression in MAIT17 cells. Moreover, we recommend to analyze cytokine expression by Rictor deficient MAIT cells (*IFN γ* , *TNF α* , *Granzyme B*, *IL-17A*, *IL-17F*, *IL-22*, *IL-21*, *IL-10* and *GM-CSF*).

Response: We have added new data showing that stimulation of ICOS can induce mTORC2 activation in MAIT cells (Figure 5C). To address the TCR signaling issue, we have analyzed mice that overexpress either WT diacylglycerol kinase ζ (DGK ζ^{wt}) or a gain of function mutant of DGK ζ (DGK $\zeta^{\Delta NLS}$) in developing thymocytes. Elevated DGK ζ activity inhibits TCR induced diacylglycerol (DAG) signaling. We have recently reported that DGK $\zeta^{\Delta NLS}$ exerted greater ability to inhibit MAIT cell maturation (*Pan et al Eur J Immunol. 2019 Nov 11. doi: 10.1002/eji.201948289.*). We have now found that while DGK ζ^{wt} did not obviously affect MAIT1/17 ratios but DGK $\zeta^{\Delta NLS}$ inhibited MAIT17 cell development more obviously than MAIT1 cell development without obviously affect ICOS expression. These data suggest that TCR signaling could affect MAIT effector lineage differentiation. These data are added to Supplemental Figure S1D, S1E, and S1F. Because Rictor deficient mice have very few MAIT cells, the assessment of cytokine expression is interesting but are not trivial experiments. However, we did include IL17A and IFN γ data (Figure 7J). We regret that we could not generate additional data to confidently measure other cytokines.

7. In Figure 7 (new figure 8 and 9), authors tried to address the role of IL1 β and IL-23 in MAIT cell biology. Based on results in Figure 7I, IL1 β signaling but not IL-23 can lead to development of double-positive T-bet⁺ROR γ t⁺ MAIT cell population, suggesting the role of IL-1 β in MAIT17 and MAIT1 cell plasticity. We suggest to answer this question by activating sorted MAIT1 or MAIT17 cells with IL-1 β . In addition it will be interesting to determine the role of IL-12 for MAIT1 development and TGF β , IL-6, and IL-21 for MAIT17 cell differentiation as well as to check the expression level of receptors for IL-21, IL-6, and IL-23 cytokines.

Response: We thank the reviewer again for suggesting many experiments to address multiple interesting questions. We did additional experiments to determine whether IL1 β and IL23 can exert significant impact on MAIT cells in vivo. We have confirmed that IL1 β and IL23 synergistically promote MAIT cell expansion in vivo. We have also found that IL1 β and IL23 induce the T-bet⁺ROR γ t⁺ double positive MAIT cells in vivo. Because these cells are ICOS⁺CD122⁻, suggesting that they are derived from MAIT17 cells. These data are shown in Figure 8I – 8K and Figure 9E – 9H. We also sorted ICOS⁺CD122⁻ MAIT17 and ICOS⁻CD122⁺ MAIT1 cells and then stimulate these cells in vitro with IL1 β and IL23. However, we are not able to obtain consistent results. We agree that plasticity is an interesting issue and it would be best address using T-bet and ROR γ t reporter mice in the future.

Minor concerns:

1. In Figure 6G please include the percentage for the ICOS⁺ population.

Response. We have added the percentages. Sorry for the error.

2. In Introduction, we recommend to extend the Introduction by including the following key aspects of MAIT cell biology: the limited array of TCR β -chain, TCR-dependent MAIT cell activation through the TLR pathway, and the cytokine profile of MAIT cells.

Response. We have included these aspects of MAIT cell biology in the Introduction or discussion.

Reviewer #2 (MAIT, MR1, T cell biology):

General comments:

Using a number of genetically modified mice and an in vitro MAIT cell activation assay, the authors firstly identified CD122 and ICOS as potential surrogate cell surface markers to define MAIT cell phenotypes: MAIT1 and MAIT17 respectively. A functional role for CD122 and ICOS on MAIT cell differentiation was also observed. Secondly, signaling via IL15R, IL1betaR and IL23R were found important for MAIT cell proliferation and phenotypic differentiation (MAIT1 and MAIT17). The authors then examined the downstream signaling components: Raptor and Rictor in the mTOR complex that shape the MAIT1 and MAIT17 polarization. The authors concluded that mTORC2 integrates signals from ICOS and IL1 β R/IL23R to exert a crucial role for MAIT17 differentiation and an IL15R-mTORC1-T-bet axis ensures MAIT1 differentiation.

The design of the experiments is appropriate to the aims but in some instances amplifies the reported findings of others, rather than representing breakthroughs. For example, recent reports by the Lantz group described CD122 as a phenotypic marker (Legoux et al., Nature immunology 2019 20(9):1244-1255) and Legoux et al., Science 10.1126/science.aaw2719 (2019). The transcriptional upregulation of IL15 is flagged in the papers in Cell Reports (2019) from the Klenerman and McCluskey groups where Leng et al also describe the importance of IL-15 in expanding MAIT1 cells in vitro. A role for IL15 in MAIT1 regulation has also reported by Sattler et al, Eur J Immunol. 2015 Aug;45(8):2286-98.

Several groups at the recent Oxford conference on CD1-MR1 reported a key role for ICOS-ICOSL interactions in MAIT cell development and activation though these are not yet published.

As acknowledged by the authors, the function of the mTOR complex on differentiation towards MAIT1 and MAIT17 has been well described in NKT1 and NKT17 differentiation (Wei, Yang, & Chi, 2014) despite the finding is novel in the field (MAIT biology).

The data set from mTOR conditional KO mice is novel in the field (MAIT biology) and will fill a knowledge gap.

Response: We regret that we missed these references and we thank the reviewer for the expertise and encouragement. We have discussed these issues and included these references in the revised manuscript.

1. Some findings may be nonspecific (and not physiological), therefore should be validated in vivo. The observations on the role of IL1beta and IL23 from in vitro experiments (Figure 7) may not be physiological and it is very important that this is verified by in vivo experiments. For instance, a critical activation role of IL18 has been reported in in vitro studies but could not be reproduced by in vivo experiments (Jesteadt et al., 2018).

Response: We have done additional experiments to determine whether IL1 β and IL23 can exert significant impact on MAIT cells in vivo. We have confirmed that IL1 β and IL23 synergistically promote MAIT cell expansion in vivo. These data are shown in Figure 8I – 8K and Figure 9E – 9H. During our revision of our manuscript, Dr. Belkaid's group has reported that IL18 plays an important role for bacterial induced cutaneous MAIT cell expansion in vivo (Constantinides et al 2019).

2. For Figure 4L: A similar study on iNKT cells, published in PNAS (Shin et al 2014) from Zhong lab, showed that although mTOR has no effect on total expression of PLZF, it controls the nuclear localization of PLZF, which is essential for iNKT-cell lineage development and effector function. The authors should ascertain whether this also occurs in MAIT cells.

Response. We thank the reviewer for raising this issue. Unfortunately, mTOR/mTORC1 deficient mice have extremely low numbers of MAIT cells. We could not get enough cells to confidently address this issue. We added discuss of this issue in the result section.

Minor issues:

1. It is important to show the total MAIT cell numbers in combination with MAIT cell frequency (%), as there seemed an unstated assumption of a binary composition of MAIT cells into (MAIT1 and MAIT 17). In reality, there is a significant proportion of MAIT cells with mixed transcriptional and functional differentiation. Notably too, the frequency and/or absolute number of MAIT1 versus MAIT17 cells may be not suggest identical shifts in some situations where other cell types are shifting as well.

Figure 2J showed that total MAIT cell number decreased in several organs in IL2/15Rb KO mice and Figure 2L shows that the MAIT17 frequency was not altered in these organs except for spleen. Did these observations indicate that absolute MAIT17 number decreased as well in this KO mice? Also, Figure 2B showed a significant increase of MAIT17(CD122-) in culture in the presence of IL15. Therefore, IL15 signals are predicted to affect MAIT17 cells, in addition to MAIT1 cells (the main point)?

Response: MAIT17 cell numbers were also decreased in IL2/15RbKO mice. We have included the data in new figure 3F. Thus, IL2/15R may also play a weak role for MAIT17 cells. IL15 exerted strong effects on MAIT1 cell expansion in vivo and in vitro but it was able to slightly induce MAIT17 expansion in vitro but not in vivo. In addition, our new data showed that IL2 also preferentially acted on MAIT1 cells to induce their expansion. However, in mixed chimeric mice, MAIT1 ratio was severely decreased but MAIT17 ratio was increased. Overall, IL2/15R are critical for MAIT1 differentiation/maintenance and may exert a weak role for MAIT17 cells.

- Figure 5G and 5H have the same issue as Figure 2L. Absolute number of MAIT17 cells should be shown as well.

Response: We have added numbers as advised. These data are shown in Figure 6G and 6H.

2. Figure 4I: this lacks a negative control from MR1^{-/-} mice. Also, in the section of materials and methods the authors did not mention which anti-MR1 antibody was used.

Response: We regret that we have not obtained MR1^{-/-} mice. However, we repeated the experiment with an isotype antibody control (Figure 5J). The new data support that MR1 expression is not decreased in CD4⁺CD8⁺ thymocytes in mTOR^{f/f}-CD4Cre mice. We have included the anti-MR1 antibody information in the method section.

3. Some citations are not accurate. For instance, Line 64 “...display cytotoxic activity^{10, 23, 24}”. Reference 10 did not mention granzyme B or cytotoxic activity.

Response: We have removed reference 10 as advised.

4. The referencing is missing some key papers, e.g. Chen et al Mucosal Immunology 10(1):58-68 describes the MAIT1/MAIT17 polarisation in vivo during infectious challenge.

Response: We regret for missing several important papers. We have referenced this paper and several other papers.

Reviewers' comments:

Reviewer #1 (Remarks to the Author):

In the revised paper the authors have addressed several concerns raised by the reviewers, but some major and critical issues are still needed to be adjusted.

Figure 1. Though the authors have included new data to define the key transcriptional factors for MAIT1 and MAIT17 cells, the newly provided scRNAseq plot (Figure 1G) has to be clarified. In particular, the shown cell populations are defined just by numbers, we suggest including a descriptive material. Moreover, it remains unclear what gating strategy (markers) did authors use to define the MAIT cell population in scRNAseq analysis, for consistency we suggest using the same strategy as was defined in Figure 1 A-E. In addition, we still insist on determining the cytokine signature for both MAIT cell populations by trying other methods besides flow cytometry staining.

Figure 2. The authors have included new data with IL-2 contribution and admitted its impact on MAIT 1 cell expansion. We suggest showing the synergy of IL-2 and IL-15 pathways for MAIT1 development by utilizing the same activation approach but with the usage of both IL-2 or IL-15 KO mice and show if the expansion may be abrogated due to lack of one of these factors.

Figure 3 (new Figure 4). The concern of ICOS-L impact on MAIT 17 lineage commitment has not been addressed thoroughly. It seems that the source of ICOS-L for MAIT cell expansion is tissue-dependable (Figure 4 J), we still encourage the authors to define the possible contribution from cells expressing ICOS-L (Macrophages, DCs, B cells, etc) and show in what time point it may be required to MAIT 17 lineage commitment.

Figure 4. We suggest showing the possible ICOS-Rictor axis in MAIT 17 development.

Figure 5 (new Figure 6). We still recommend showing the synergy between IL-2 and IL-15 in promoting MAIT 1 cells by including 1 more condition (IL-2 + IL-15) in Figure 6 J.

Figure 6. We had asked to show crosstalk between mTORc2 and ICOS, however, the authors didn't demonstrate earnestly either the existing axis or possible crosstalk. We recommend utilizing the overexpression method to show if it can rescue the deficiency or not. Additionally, we still want the authors to determine the role of TCR signaling in ICOS expression in MAIT17 cells since the provided data didn't answer this question.

Figure 7 (new figures 8 and 9). In newly included data the authors have determined the T-bet+RORy-t+ double-positive MAIT cell population. We encourage demonstrating the possible role of this population and discuss the path of its development (it has not been detected before in thymic samples). In addition, we suggested showing IL-1b contribution for MAIT 17 development more precisely. The shown in vitro experiment was done with splenocytes. To clearly define the role of either IL-23 or IL-1b or both of them we strongly encourage the authors to perform clean in vitro experiment with sorted MAIT cells. Moreover, in Figure 9E we question the ratio and percentage of single-positive populations for T-bet+ and RORy-t+ since in previously shown figures (Figure 1C) the majority of cells were RORy-t+.

Reviewer #2 (Remarks to the Author):

The revised manuscript is significantly improved in quality. The novelty of the manuscript remains the major issue notwithstanding the originality of the mTOR data.

While the revised manuscript has addressed one of the two major concerns, the main conclusion on the role of IL-23 and the importance of ICOS on MAIT cell activation (Fig 8 and 9), remains a modest advance on what is known from the 2019 CD1-MR1 conference in Oxford and the published work from

Wang et al, Science Immunology 2019.

The second major concern raised in the initial review was not addressed for technical reasons.

Re: minor concerns:

1. In Figure 5J, MR1^{-/-} mice were sought as negative controls but they have not been included in the revised Ms. It is known that anti-MR1 antibodies (26.5 and 8F2) do not function well in FACS staining of primary cells from B6 vs MR1^{-/-} mice. This is true in wild type and infected mice, although these mAbs can be used for blocking purpose. Hence, the data addressing this issue are not convincing.
2. In Fig. 6G and 6H, it is recommended that the authors check the statistics used in evaluating the MAIT17 numbers noting the visual discrepancy in the right hand panels. Control mice possessed many more MAIT17 cells visually than their KO counterparts.
3. It would be valuable to see in vivo data relating to Figure 9E-G showing the inhibitory effect of rapamycin on IL b and IL 23 as demonstrated in Fig 6J

Point-by-point responses to Reviewers' comments:

In the revised manuscript, we have made the following changes.

- 1). Addition of Supplemental figure S1D. Previous supplemental Figures S1D, S1E, and S1F are changed to Figure S1E, S1F, and S1G, respectively.
- 2). Addition of Supplemental Figure S2, which shows differential gene expression in MAIT cell clusters. Previous supplemental Figure S2 – S9 are changed to Figure S3 – S10, respectively.
- 3). Addition of supplemental figure S5A and S5B. Previous supplemental Figure S4A – S4D (now Figure S5) are changed to S5C – S5F respectively.
- 4). Deleted previous Figure 5J, which is a overlaid histogram showing Mr1 expression in DP thymocytes from WT and mTOR-Cd4Cre mice.
- 5). Included rapamycin inhibition data for Figure 9B and 9C.

Reviewer #1:

In the revised paper the authors have addressed several concerns raised by the reviewers, but some major and critical issues are still needed to be adjusted.

Figure 1. Though the authors have included new data to define the key transcriptional factors for MAIT1 and MAIT17 cells, the newly provided scRNAseq plot (Figure 1G) has to be clarified. In particular, the shown cell populations are defined just by numbers, we suggest including a descriptive material. Moreover, it remains unclear what gating strategy (markers) did authors use to define the MAIT cell population in scRNAseq analysis, for consistency we suggest using the same strategy as was defined in Figure 1 A-E. In addition, we still insist on determining the cytokine signature for both MAIT cell populations by trying other methods besides flow cytometry staining.

Response: We have included additional description of the cell populations of scRNA analysis as advised. The scRNAseq data analysis was done using the unsupervised principle component analysis and k-nearest neighbors analysis. We are sorry that we are not able to perform the analysis using the gating strategy suggested by the reviewer. However, clusters identified using the unsupervised approach match wells with the MAIT1 and MAIT17 cells we have identified by FACS. We have also sorted liver CD122⁺ICOS⁻ and CD122⁻ICOS⁺ MAIT cells and used a multiplex approach to cytokines in these cells. We have found that CD122⁺ICOS⁻ MAIT cells produced IFN γ , IL2, and TNF α but not IL17A while CD122⁻ICOS⁺ MAIT cells produced IL17A, IL2, and TNF α but lower concentration of IFN γ . The new data are shown in supplemental Figure S1D. This is not a trivial experiment as it requires to use many mice to sort enough MAIT cells and needs multiple optimizations for detecting cytokines in culture supernatants.

Figure 2. The authors have included new data with IL-2 contribution and admitted its impact on MAIT 1 cell expansion. We suggest showing the synergy of IL-2 and IL-15 pathways for MAIT1 development by utilizing the same activation approach but with the usage of both IL-2 or IL-15 KO mice and show if the expansion may be abrogated due to lack of one of these factors.

Response: We have examined whether IL2 and IL15 synergize to induce MAIT cell expansion as advised. The new data are shown in supplemental Figure S5. It appears that injection of IL15/IL15R α FC complexes induced MAIT1 cell expansion slightly better than IL2/anti-IL2 antibody complexes. However, we do not see obvious synergy between these two cytokines in inducing WT MAIT1 expansion. The new data are shown in supplemental Figure S5A and S5B). We did not test IL-2 or IL-15 KO mice to detect MAIT1 cell expansion as we do not have the mice and establish the colonies in the lab will take long-time and we are extremely limited by available funding and man power to start another two lines of mice in the lab.

Figure 3 (new Figure 4). The concern of ICOS-L impact on MAIT 17 lineage commitment has not been addressed thoroughly. It seems that the source of ICOS-L for MAIT cell expansion is tissue-dependable (Figure 4 J), we still encourage the authors to define the possible contribution from cells expressing ICOS-L (Macrophages, DCs, B cells, etc) and show in what time point it may be required to MAIT 17 lineage commitment.

Response: we agree with the reviewer that the source of ICOS-L directing MAIT17 lineage commitment is an interesting issue. Given its importance, we believe that it would require conditional ablation of ICOS-L in specific lineages to provide informative data to define its role in different cell lineages for MAIT17 development. We hope that the reviewer would agree with us that this is a future direction we can investigate in the future.

Figure 4. We suggest showing the possible ICOS-Rictor axis in MAIT 17 development.

Response: Our data indicate that ICOS deficiency and Rictor deficiency preferentially affects MAIT17 cells and that ICOS stimulation can induce mTORC2 activation. These data support that an ICOS-mTORC2 pathway to promote MAIT17 differentiation. We agree there are more work to be done to further analyze this pathway. We hope that the reviewer would agree that the data we have presented support this pathway.

Figure 5 (new Figure 6). We still recommend showing the synergy between IL-2 and IL-15 in promoting MAIT 1 cells by including 1 more condition (IL-2 + IL-15) in Figure 6 J.

Response: We did not observe obvious synergy between IL2 and IL-15 for MAIT1 expansion or T-bet upregulation in MAIT1 cells, shown in new supplemental figure S5A and S5B.

Figure 6. We had asked to show crosstalk between mTORc2 and ICOS, however, the authors didn't demonstrate earnestly either the existing axis or possible crosstalk. We recommend utilizing the overexpression method to show if it can rescue the deficiency or not. Additionally, we still want the authors to determine the role of TCR signaling in ICOS expression in MAIT17 cells since the provided data didn't answer this question.

Response: During the initial revision, we have added experiments showing that stimulation of ICOS with an agonist antibody induced mTORC1 and mTORC2 activation in CD122⁺ MAIT1 and CD122⁻ mostly MAIT17 cells (Figure 5C). Additionally, we showed ICOS expression was decreased in both stage 2 and stage 3 MAIT cells (Figure 7O), suggesting that mTORC2 was involved in upregulation of ICOS expression in MAIT cells to promote MAIT17 differentiation. In supplemental Figure S1G, we showed that downregulation of diacylglycerol mediated signaling in DGK ζ DNLS mutant mice caused decreases of MAIT17 cells without obviously affect ICOS levels in these cells, suggesting that at least DAG-mediated

signaling might not be critical for ICOS expression. We appreciate the reviewer's suggestion of overexpression methods for rescue. Unfortunately, we do not have mice overexpressing of ICOS or Rictor. We have tested retroviral mediated transfer of genes into bone marrow hemopoietic stem cells and generate bone marrow chimeric mice. However, the low transduction efficiency coupled with extremely low MAIT cell percentages makes it hard to generate reliable data. Additionally, Rictor coding sequence is too big to be packed into retrovirus. We hope that the reviewer would agree with us this issue can be studied in the future.

Figure 7 (new figures 8 and 9). In newly included data the authors have determined the T-bet⁺RORγ^{t+} double-positive MAIT cell population. We encourage demonstrating the possible role of this population and discuss the path of its development (it has not been detected before in thymic samples). In addition, we suggested showing IL-1β contribution for MAIT 17 development more precisely. The shown in vitro experiment was done with splenocytes. To clearly define the role of either IL-23 or IL-1β or both of them we strongly encourage the authors to perform clean in vitro experiment with sorted MAIT cells. Moreover, in Figure 9E we question the ratio and percentage of single-positive populations for T-bet⁺ and RORγ^{t+} since in previously shown figures (Figure 1C) the majority of cells were RORγ^{t+}.

Response: We thank the reviewer for pointing out the potential importance of the T-bet⁺RORγ^{t+} MAIT cells induced by IL1β and IL23. At present, we do not know the developmental path of this population of cells. We sorted CD122⁺ICOS⁻ and CD122⁻ICOS⁺ MAIT cells and examined whether they displayed different potential to become T-bet⁺RORγ^{t+} double positive MAIT cells. Unfortunately, we did not obtain conclusive data from the experiments. We would greatly appreciate if the reviewer would allow us to study this important issue in depth in the future. We have added discussion about this population in the discussion as advised.

Figure 1C showed thymus MAIT cells, which contain predominantly RORγ^{t+} MAIT17 cells but very few T-bet⁺ MAIT1 cells. However, about 50% of liver MAIT cells are MAIT1 cells (Figure 1I). For Figure 9E and 9F, we examined liver MAIT cells. The averages T-bet⁺RORγ^{t-} and T-bet⁺RORγ^{t+} MAIT cells in PBS injected (ctrl) group were similar to those shown in Figure 1I. Figure 9E represents one experiment.

Reviewer #2 (Remarks to the Author):

The revised manuscript is significantly improved in quality. The novelty of the manuscript remains the major issue notwithstanding the originality of the mTOR data.

While the revised manuscript has addressed one of the two major concerns, the main conclusion on the role of IL-23 and the importance of ICOS on MAIT cell activation (Fig 8 and 9), remains a modest advance on what is known from the 2019 CD1-MR1 conference in Oxford and the published work from Wang et al, Science Immunology 2019.

Response. We thank the reviewer for the comment that we have significantly improved the quality of the manuscript. With regard to the role of ICOS, our findings that ICOS promotes mTORC2 activation in MAIT cells and is important for MAIT17 lineage differentiation are new and have not been previously published.

The second major concern raised in the initial review was not addressed for technical reasons.

Response: We thank the reviewer for the understanding.

Re: minor concerns:

1. In Figure 5J, MR1^{-/-} mice were sought as negative controls but they have not been included in the revised Ms. It is known that anti-MR1 antibodies (26.5 and 8F2) do not function well in FACS staining of primary cells from B6 vs MR1^{-/-} mice. This is true in wild type and infected mice, although these mAbs can be used for blocking purpose. Hence, the data addressing this issue are not convincing.

Response: We thank the reviewer's expertise on this issue. We have removed Figure 5J from the manuscript as the data is not necessary for our conclusion that mTOR intrinsically control MAIT cell development based on the data in mixed bone marrow chimeric mice.

2. In Fig. 6G and 6H, it is recommended that the authors check the statistics used in evaluating the MAIT17 numbers noting the visual discrepancy in the right hand panels. Control mice possessed many more MAIT17 cells visually than their KO counterparts.

Response: We have checked the data as advised. The data are correct. Because there are rptorKO mice that have not MAIT cells, we could not use log scale for the Y-axis.

3. It would be valuable to see in vivo data relating to Figure 9E-G showing the inhibitory effect of rapamycin on IL 1b and IL 23 as demonstrated in Fig 6J.

Response: We thank the reviewer for raising this interesting issue. We actually tested the effect of rapamycin on IL1 β and/or IL23 induced MAIT cell expansion in vitro but did not include the data in previous revision. We are now including the data in Figure 9B and 9C. Although we did not test the effects of rapamycin in vivo, the in vitro experiment indicates that rapamycin greatly inhibited IL1 β /IL23 induced expansion of T-bet⁺ROR γ t⁻, T-bet⁺ROR γ t⁺, and T-bet⁻ROR γ t⁺ MAIT cells. The role of IL1 β signal in MAIT cells is very interesting. We plan to further investigate this using IL1 receptor deficient mice in the future.

REVIEWER COMMENTS

Reviewer #1 (Remarks to the Author):

In the revised manuscript, the authors by performing additional experiments have addressed several concerns that were raised during the second round of revision. We agree with the authors' explanation that some experiments require tremendous time, labor force and sources to answer the proposed questions. Nevertheless, there are several points, which must be addressed prior to publication.

Figure 1. Though the authors have performed the multiplex analysis to dissect the range of MAIT cell cytokine expression, we would like to have an explanation of the chosen source of MAIT cells for the cytokine analysis. Throughout the whole manuscript, the authors have been investigating MAIT cells from different tissues such as spleen, pLNs, lung and liver; thus, we recommend either performing the same multiplex assay for all tested tissues or explain why liver was chosen as the source to sort MAIT cells.

Figure 7. We still encourage the authors to perform "clean" in vitro experiment to determine the impact of IL-1b and IL-23 cytokines on MAIT cells expansion. Specifically, instead of utilizing splenocytes we recommend sorting out MAIT cells and performing the same in vitro cytokines activation assay with the clean MAIT cell population.

Point-by-point responses to reviewers' comments

Reviewer #1

In the revised manuscript, the authors by performing additional experiments have addressed several concerns that were raised during the second round of revision. We agree with the authors' explanation that some experiments require tremendous time, labor force and sources to answer the proposed questions. Nevertheless, there are several points, which must be addressed prior to publication.

Comment 1. Figure 1. Though the authors have performed the multiplex analysis to dissect the range of MAIT cell cytokine expression, we would like to have an explanation of the chosen source of MAIT cells for the cytokine analysis. Throughout the whole manuscript, the authors have been investigating MAIT cells from different tissues such as spleen, pLNs, lung and liver; thus, we recommend either performing the same multiplex assay for all tested tissues or explain why liver was chosen as the source to sort MAIT cells.

Response: We are sorry that we are not able to analyze more tissues with multiplex cytokine assays. It is very difficult to obtain enough MAIT cells, especially MAIT1 cells, for measuring cytokines by multiplex. We sort liver MAIT cells to detect cytokines because liver has close to equal numbers of MAIT1 and MAIT17 cells. Other organs contain predominantly MAIT17 cells and very few MAIT1 cells. We tried our best to examine different tissues with different mouse models to provide a more complete pictures in our manuscript.

Comment 2. Figure 7 (Figure 8). We still encourage the authors to perform “clean” in vitro experiment to determine the impact of IL-1b and IL-23 cytokines on MAIT cells expansion. Specifically, instead of utilizing splenocytes we recommend sorting out MAIT cells and performing the same in vitro cytokines activation assay with the clean MAIT cell population.

Response: We have performed experiments as advised. The new data show that IL1 β and IL23 induce MAIT17 cell but not MAIT1 cell expansion in vitro. Additionally, we also show that MAIT17 cells but not MAIT1 cells can turn into ROR γ t+T-bet+ cells in this in vitro setting. These new data are shown in Figure 9I.

Reviewer 2.

We have analyzed two additional pairs of RaptorKO mice and have updated the data in Figure 6. We have also examined the effects of rapamycin on IL1 β +IL23 induced MAIT cell expansion in vivo and show that inhibition of mTOR by rapamycin substantially reduces MAIT cell expansion in vivo (Figure 8L). These data address concerns of Reviewer 2.

REVIEWERS' COMMENTS

Reviewer #1 (Remarks to the Author):

During the final round of the revision, the authors have addressed all raised concerns. We have evaluated the Figure 9I and agreed that the impact of IL-1b and IL-23 cytokines on MAIT cells was discussed accurately. Regarding the Figure 1, we understand that the sorting of certain cell populations is an arduous process and believe that liver derived MAIT cells are sufficient to perform multiplex assay for measuring the cytokine level.

Point-by-point responses to reviewers' comments

Reviewer #1 (Remarks to the Author):

During the final round of the revision, the authors have addressed all raised concerns. We have evaluated the Figure 9I and agreed that the impact of IL-1b and IL-23 cytokines on MAIT cells was discussed accurately. Regarding the Figure 1, we understand that the sorting of certain cell populations is an arduous process and believe that liver derived MAIT cells are sufficient to perform multiplex assay for measuring the cytokine level.

Response: We greatly appreciate the reviewer's expertise and many insightful comments during the reviewing process.